

# An information content approach to diagnosing and improving CLIMCAPS retrievals across instruments and satellites

Nadia Smith[1], Chris D. Barnet[1]

[1]Science and Technology Corporation, Columbia, MD, 21046, United States

*Correspondence to*: Nadia Smith (nadias@stcnet.com)

**Abstract.** The Community Long-term Infrared Microwave Combined Atmospheric Product System (CLIMCAPS) characterizes the atmospheric state as vertical profiles (commonly known as soundings or retrievals) of temperature, water vapor, $CO_2$, CO, $CH_4$, $O_3$, $HNO_3$ and $N_2O$, together with a suite of Earth surface and cloud properties. The CLIMCAPS record spans more than two decades (2002–present) because it utilizes measurements from a series of different instruments on different satellite platforms. Most notably, these are AIRS+AMSU (Atmospheric Infrared Sounder + Advanced Microwave Sounding Unit) on Aqua and CrIS+ATMS (Cross-track Infrared Sounder + Advanced Thermal Microwave Sounder) on SNPP and the JPSS series. Both instrument suites are on satellite platforms in low-Earth orbit with local overpass times of ~1:30 am/pm. The CrIS interferometers are identical across the different platforms, but differ from AIRS, which is a grating spectrometer. At first order, CrIS+ATMS and AIRS+AMSU are similar enough to allow a continuous CLIMCAPS record, which was first released in 2020 as Version 2 (V2). In this paper, we take a closer look at CLIMCAPS V2 soundings from AIRS+AMSU (on Aqua) and CrIS+ATMS (on SNPP) to diagnose product continuity across the two instrument suites. We demonstrate how averaging kernels, as signal-to-noise ratio (SNR) indicators, can be used to understand and improve multi-instrument systems such as CLIMCAPS. We conclude with recommendations for future CLIMCAPS upgrades.

## 1 Introduction

The launch of Aqua on 4 May 2002 heralded in a new era for satellite sounding of the Earth atmosphere with the Atmospheric Infrared Sounder (AIRS) onboard Aqua (Aumann et al., 2003; Chahine et al., 2006). The National Aeronautics and Space Administration (NASA) AIRS instrument was the first of its kind in space, and measures emitted infrared (IR) radiance with hundreds of spectrally narrow channels and low instrument noise. Its high spectral resolution allows AIRS to measure a wide range of parameters about the thermodynamic structure *and* chemical composition of the atmosphere (Susskind et al., 2003). More than 21 years later and still in operational orbit, Aqua continues to contribute important Earth system measurements (Parkinson, 2003, 2013, 2022). But, with only a few years before the spacecraft is decommissioned, it becomes important to develop ways that would see the Aqua record continue with next-generation instruments and platforms.





The Community Long-term Infrared Microwave Combined Atmospheric Product System (CLIMCAPS) retrieves atmospheric soundings from hyperspectral infrared (IR) measurements, like those made by AIRS. CLIMCAPS soundings are profiles of temperature, water vapor and a host of minor gases ($CO_2$, $CO$, $CH_4$, $O_3$, $N_2O$, $HNO_3$) as well as cloud and Earth surface properties (Smith and Barnet, 2023a). CLIMCAPS augments measurements from AIRS with those from AMSU (Advanced Microwave Sounding Unit), which is also on the Aqua spacecraft. AIRS provides the bulk of the information about the

atmospheric state, while AMSU helps distinguish whether a retrieval scene is uniformly clear or uniformly cloudy. CLIMCAPS builds on decades of NASA and NOAA (National Oceanic and Atmospheric Administration) investment in sounding science, and extends the Aqua record with soundings from CrIS+ATMS on the Suomi National Polar-orbiting Partnership (SNPP) and Joint Polar Satellite System series (JPSS-1 to JPSS-4). The AIRS Science Team (AST) applied their retrieval method to CrIS+ATMS on SNPP (Susskind et al., 2013, 2017), but it was not until CLIMCAPS explicitly addressed

instrument differences in its retrieval approach that a continuous record between AIRS+AMSU and CrIS+ATMS was established (Smith and Barnet, 2023a). We use the term "continuous" to mean a data record that is consistent in its characterization of natural variation despite changes in source instrumentation.

    There are two other instruments on Aqua still contributing valuable measurements and that face their own set of challenges as

far as the continuation of their data records go; these are the Clouds and the Earth's Radiant Energy System (CERES; Barkstrom, 1990; Wielicki et al., 1996) as well as the MODerate resolution Imaging Spectroradiometer (MODIS; Barnes and Salomonson, 1992). Continuity of the Aqua/CERES data record is, perhaps, the most straightforward. Unlike AIRS and MODIS, the CERES instrument design is largely unchanged across the various payloads spanning almost three decades (Loeb et al., 2018). Instrument design alone, however, does not ensure data continuity, which is demonstrated by CERES science

team efforts to overcome in-orbit variation in calibration (Loeb et al., 2016; Shankar et al., 2023) as well as inconsistencies in ancillary datasets (Kato et al., 2018; Su et al., 2020). The continuity of MODIS with measurements from the Visible Infrared Imaging Radiometer Suite (VIIRS) is complicated by differences in spectral coverage and spatial resolution that affect the quality and continuity of a wide range of long-term records. These include NASA products characterizing land properties (Román et al., 2024), sea-surface temperature (Minnett et al., 2020), snow and ice trends (Riggs et al., 2017; Riggs and Hall,

2020), as well as the optical and spatial properties of clouds (Platnick et al., 2020; Meyer et al., 2020, respectively). Some approaches to address instrument differences led to new, innovative products such as the imager-sounder fusion technique that fills VIIRS spectral gaps to better mimic MODIS measurements (Borbas et al., 2021; Weisz et al., 2017), and the Climate Hyperspectral Infrared Radiance Product (CHIRP) that derives a set of long-term radiance spectra drawing on commonalities between AIRS and CrIS (Strow et al., 2021).


    In this paper, we focus our attention on diagnosing the CLIMCAPS V2 record (Barnet, 2019; Sounder SIPS and Barnet, 2020a, b) to evaluate data continuity across the different instrument suites and determine how it can be improved. The CLIMCAPS retrieval approach has its origin in the NASA AST retrieval method, which deviates from traditional Bayesian Optimal



Estimation (OE, Rodgers 2000) in a number of ways. Most notably, the NASA AST method employs dynamic, scene-dependent regularization with singular value decomposition (SVD) to the signal-to-noise ratio (SNR) matrix (see Smith and Barnet, 2020 for details). The AST method was originally developed (Susskind et al., 2003) and later optimized (Susskind et al., 2014) for the retrieval of atmospheric soundings from AIRS+AMSU. We refer to this product as "AST-Aqua V7" from here on. With the launch of similar instruments – IASI+AMSU (Infrared Atmospheric Sounding Interferometer + Advanced Microwave Sounding Unit) on MetOp in 2006 and the CrIS+ATMS on Suomi-NPP (National Polar-orbiting Partnership) in 2007 and the Joint Polar Satellite System (JPSS+) series in 2011, 2017 and 2022 – NOAA implemented the AST method as NUCAPS (NOAA-Unique Combined Atmospheric Processing System) to retrieve satellite soundings from different times of the day (Barnet et al., 2021). IASI+AMSU measure the atmosphere ~9:30 am/pm and CrIS+ATMS ~1:30 am/pm. At present, however, NOAA does not synchronize NUCAPS upgrades across all CrIS+ATMS and IASI+AMSU configurations, nor do they reprocess the full record whenever an algorithm change is introduced, so NUCAPS soundings do not readily characterize atmospheric change over time. Instead, NOAA optimizes NUCAPS products to be available in *real-time* (i.e., within ~60 min of satellite overpass) to the National Weather Service (NWS) in support of severe weather forecasting (Berndt et al., 2020; Esmaili et al., 2020; Smith et al., 2018; Weaver et al., 2019). CLIMCAPS, in contrast, is designed to be synchronized across multiple instrument configurations for the depiction of atmospheric change over days, months and years. In support of this, GES DISC (Goddard Earth Sciences Data and Information Services Center) reprocesses the full CLIMCAPS record with every system upgrade to enable the study of atmospheric processes and their seasonal and intra-annual variation. The CLIMCAPS retrieval algorithm itself is very fast; on the order of 0.27s per sounding retrieval per central processing unit, or CPU (Smith and Barnet 2023a). This means that the full record of observations (324,000 per day per satellite) can be reprocessed with modest computational resources.

It is worth distinguishing between the CLIMCAPS V2 product as it is currently available at GES DISC, and the CLIMCAPS retrieval system in general. The CLIMCAPS algorithm design reflects decades of NASA investment in sounder science and retrieval theory with its ability to support experimentation and instrument innovation. In one instance, we emulated NUCAPS capability by running an experimental configuration of CLIMCAPS on AIRS+AMSU measurements. Paired with NOAA soundings from NUCAPS using CrIS+ATMS measurements, forecasters had access to a time-series of sounding observations for improved situational awareness during pre-convective forecasting (Berndt et al., 2023). CLIMCAPS and NUCAPS transform each swath of instrument measurements into a three-dimensional (3-D) depiction of the instantaneous atmospheric state. By paring sounding retrievals from different instrument suites with measurements at different times, one can obtain 3-D information about atmospheric change that benefit severe weather forecasting in novel ways (Berndt et al., 2020; Esmaili et al., 2020; Smith et al., 2018, 2019; Wheeler et al., 2018) and support the development of innovative products, such as motion vectors as proxy for wind measurements (Ouyed et al., 2023). In another instance, we ran the CLIMCAPS system on subsets of short-wave IR channels to test the fecundity of instruments with limited spectral band capability (Barnet et al., 2023). CLIMCAPS is a system that incorporates the knowledge and community contribution from both NOAA and NASA. It has the



ability to employ any number of a-priori estimates, whether from reanalyses, forecast models or different regression retrievals. Moreover, we can run CLIMCAPS on different channel subsets, variation in cloud clearing or the order of retrieval steps.
CLIMCAPS is a mature system that can support scientific experiments with traceable error estimates and rapid operational deployment for target applications if needed.

This said, our focus in this paper is on the CLIMCAPS V2 product suite released by GES DISC in 2020, which is the only data record to date that spans the combined lifetimes of AIRS+AMSU and CrIS+ATMS with a consistent retrieval approach.
With the innovation and maturation of science applications (Gaudel et al., 2024; Ouyed et al., 2023; Prange et al., 2023; Smith et al., 2021), we revisit the CLIMCAPS V2 record to diagnose and mitigate as many systematic effects as possible for the sake of a seamless characterization of the 3-D Earth atmosphere. In Section 2, we discuss why this research is necessary and summarize the CLIMCAPS components that afford a diagnostic evaluation of its sounding products. In Section 3, we outline and justify our experimental design, and in Section 4 discuss results. Section 5 summarizes our conclusions for future system
improvements. The analysis presented in this paper uses CLIMCAPS retrievals from Suomi-NPP CrIS+ATMS and Aqua AIRS+ATMS measurements made in 2016 when both instrument suites were in operational orbit at full spectral resolution. Henceforth, we distinguish these two systems as "CLIMCAPS-Aqua" and "CLIMCAPS-SNPP". Note that a demonstration of continuity between CLIMCAPS-Aqua and CLIMCAPS-SNPP implies continuity with CLIMCAPS-JPSS+, since all four JPSS platforms, JPSS-1 through JPSS-4, have the same CrIS+ATMS instruments. This means that the CLIMCAPS sounding record
has potential to span at least four decades, 2002 to ~2040.

## 2 The CLIMCAPS retrieval approach

Unlike NUCAPS and AST-Aqua V7, CLIMCAPS is supported through a competitive NASA grant system that funds, at most, the full-time-equivalent (FTE) of one expert for three years. With such limited resources, CLIMCAPS capability can only be maintained and incrementally improved given collaborative efforts within the community at large. The full CLIMCAPS record
was made publicly available for the first time in 2020 as Version 2 (Table S1). A new grant awarded in 2021 allowed us to respond to requests from the scientific community to simplify product design (Smith et al., 2021) and improve continuity in soundings between AIRS+AMSU and CrIS+ATMS. Where the initial development of CLIMCAPS drew on decades of stove-piped efforts at NASA and NOAA, respectively, the full CLIMCAPS V2 record now in public domain allows us to make targeted upgrades using diagnostic criteria suited to known product applications. This section describes the algorithm elements
we employed to address systematic instrument differences in the CLIMCAPS V2 record. Detailed descriptions of the CLIMCAPS retrieval approach can be found elsewhere (Smith and Barnet, 2019, 2020, 2023a, 2023b.). Of interest here is the fact that CLIMCAPS can run in diagnostic mode to generate myriad quantitative metrics for in-depth evaluation of the retrieval system and a sounder science approach to algorithm upgrades.



## 2.1 Dynamic regularization using information content analysis

130 In a general sense, Bayesian optimal estimation (OE) is a method for generating quantitative information about a target feature by combining a-priori knowledge with independent measurements (Rodgers, 2000). The degree to which the measurements contribute to a final solution depends on measurement error as well as estimates of uncertainty about the a-priori. Bayesian OE yields a different solution for each combination of a-priori estimate and measurement. A Bayesian solution can, therefore, be tailored to the SNR requirements of a target application.


 Rodgers (2000) popularized OE for use in retrieving atmospheric information from infrared satellite measurements. Rodgers OE uses the a-priori error covariance matrix ($\mathbf{S}_a$) to regularize the amount of information a satellite measurement contributes to the final solution (Eq. 1 in Smith and Barnet, 2020). Such an $\mathbf{S}_a$ matrix is typically calculated off-line and applied as a static regularization term to all measurements, independent of prevailing conditions that affect measurement sensitivity to the target

140 observable. CLIMCAPS differs from Rodgers OE most notably in how it defines its regularization term. Instead of static regularization that uses a statistical estimate of a-priori uncertainty, CLIMCAPS regularizes its solution *dynamically* based on the information content in each measurement (Eq. 2 in Smith and Barnet, 2020). What this means in practice is that CLIMCAPS uses SVD at every retrieval scene to decompose the measurement SNR matrix, $\widetilde{\mathbf{K}}^\mathbf{T}\mathbf{S}_m^{-1}\widetilde{\mathbf{K}}$, into a set of orthogonal functions, where $\widetilde{\mathbf{K}}$ is the CLIMCAPS Jacobian matrix and $\mathbf{S}_m$ the measurement error covariance as defined in Smith and Barnet (2019).

145 One can interpret this set of functions as quantifying the measurement degrees-of-freedom for signal (DOFS) of the target parameter. The CLIMCAPS regularization term is a threshold value that determines the subset of strongest functions to be used in the retrieval without any regularization (or damping); for example, five undamped functions mean five pieces of information about the vertical structure of the retrieved parameter with high enough SNR to be considered independent of the a-priori estimate. All functions with eigenvalues below this threshold are damped to regularize their contribution to the retrieval

150 and thus minimize noise (see Smith and Barnet 2020 for details). The degree to which these orthogonal functions are damped – or the number of functions that are *not* damped – varies from scene to scene since both $\widetilde{\mathbf{K}}$ and $\mathbf{S}_m$ have a large dynamic range within the CLIMCAPS system (Smith and Barnet, 2019). In other words, CLIMCAPS DOFS varies according to measurement SNR at the target scene. A measurement that has a high (low) number of functions with eigenvalues greater than the threshold value, yields a retrieval with lower (higher) dependence on the a-priori across the vertical atmospheric column. This dynamic

155 regularization approach, first developed by (Susskind et al., 2003), not only stabilizes the retrieval under a wide range of atmospheric conditions, it also sharpens the vertical resolution of retrievals where appropriate.

## 2.2 Instrument-specific algorithm components

 CLIMCAPS is instrument agnostic at its core, which enables the same code to run all configurations, eliminating many
160 potential discontinuities due to version control. Instrument design directly affects measurement information content, and



therefore retrieval quality. For example, spectral coverage determines which atmospheric gases can be retrieved, while spectral resolution and instrument resolution determines their retrieval quality. Many of the Level-1 product differences are addressed and neutralized in the CLIMCAPS pre-processor, but fundamental instrument differences that affect measurement SNR propagate systematically into the retrieved quantities. As a result, instrument biases can disrupt the geophysical consistency of

a long-term data record even if a retrieval system is instrument agnostic. AIRS is a grating spectrometer (Aumann et al., 2003; Chahine et al., 2006) and CrIS an interferometer (Glumb et al., 2002; Strow et al., 2013). Smith and Barnet (2019) summarized these two instruments from a retrieval perspective and elaborate on a few key aspects in Table S1. Given these fundamental instrument differences, one has to consider AIRS and CrIS spectral resolution and how their instrument design affects the fecundity of their spectral channels. But IR spectra do not provide the only source of information. CLIMCAPS additionally

harvests spatial information from the cluster of FOVs making up each retrieval FOR, as well as spectral information from the collocated MW sounders, AMSU on Aqua and ATMS on SNPP and JPSS+. The reason for this is that retrieval parameters can be mathematically degenerate within a single source of measurement. Adding other sources can help break this degeneracy to allow the retrieval of discreet parameters (see Table 1 in Smith and Barnet, 2023a for a full list). A good example is the degeneracy of cloud and surface parameters within the IR radiances. Adding MW and spatial information to the retrieval

process helps CLIMCAPS distinguish between cloud tops and Earth surface. While the IR sounders provide the primary source of spectral information, one should always keep in mind that the spatial arrangement of IR FOVs and the quality of collocated MW measurements also affect CLIMCAPS retrieval quality.

AIRS and CrIS each have hundreds of spectral channels measuring IR radiation in narrow intervals. In CLIMCAPS, we do

not use all the available IR channels for each parameter retrieval because there is no reason to include channels insensitive to the target parameter that would be filtered out during dynamic SVD regularization anyway. So, for the sake of retrieval speed and improved SNR, we pre-select subsets of channels for each target variable as discussed in Gambacorta and Barnet (2013). The channel subsets we implemented in CLIMCAPS V2 for AIRS and CrIS, reflect best practices currently at NASA and NOAA for each instrument configuration (Table S2). But persistent differences in AKs between CLIMCAPS-Aqua and

CLIMCAPS-SNPP (Smith and Barnet, 2020), reflect systematic effects in SNR that could be traced back to CLIMCAPS V2 channel subsets. We, therefore, revisit the V2 channel sets in this paper and make recommendations for upgrades to improve the consistency in CLIMCAPS retrieval quality across AIRS+AMSU and CrIS+ATMS. Note that we select the CLIMCAPS channel sets from the full range that is available for CrIS (2211 in total). For AIRS, however, the full channel set (2378 in total) is first reduced into a "pristine" list (less than 1600 after sub-setting) ahead of channel selection to remove channels

with measurable noise effects due to, for example, large thermal cycles in orbit or on Earth (Manning et al., 2020).

### 2.3 Diagnostic metrics

CLIMCAPS yields a number of diagnostic metrics that can be used to evaluate its data record. For this study, we predominantly employed three types of metrics to analyse the degree to which retrievals from the two different instrument suites,



AIRS+AMSU and CrIS+ATMS, are consistent in space and time. These are (i) cloud clearing metrics that quantify the random and systematic error introduced by clouds, (ii) AKs and DOFS that quantify measurement SNR, and (iii) the difference between the final retrieval and a-priori estimate, which we will refer to as ADIFF from here on. AKs quantify the degree to which a solution depends on the measurement. It is a unitless quantity where *zero* means that the solution is equal to the a-priori estimate, and *one* means that the solution is entirely derived from the measurements without any contribution from the a-priori. A metric that complements the AKs, is the degree to which the solution deviates from the a-priori, i.e., ADIFF. This is an especially informative metric in CLIMCAPS where all the a-priori estimates are independent of the instrument measurements (Smith and Barnet, 2019), unlike NUCAPS and AST V7. For temperature, water vapor and $O_3$ the CLIMCAPS a-priori is dynamically defined by the reanalysis model, MERRA-2 (Gelaro et al., 2017; GMAO, 2015), that is interpolated in time and space to each instrument footprint ahead of retrieval. For the well-mixed gases, $CO_2$, $N_2O$, and $CH_4$, we use estimates of their long-term trends across seasons and hemispheres. The CLIMCAPS CO a-priori is a seasonal and inter-hemispheric climatology, and for $HNO_3$ and $SO_2$ we have a single static profile at this time. As knowledge of these gases grows, we can consider developing new climatologies and/or employing other chemistry or reanalysis models. Having a-priori estimates that are instrument independent is one of the key aspects that enable continuity in the CLIMCAPS record since they help avoid abrupt changes when new instrument measurements are introduced or when SNR is low. A CLIMCAPS retrieval typically deviates from its a-priori estimate when the measurements add new information about the true state of the atmosphere within the FOR. On rare occasions, however, a retrieval can be dominated by noise when SNR is very low. One can diagnose CLIMCAPS retrievals to better understand the reasons for ADIFF > 0 values using the AK matrices that are reported for every retrieval parameter at every FOR in the Level 2 product. These are discussed in detail in Smith and Barnet (2020) but is worth summarizing here:

(1) High AK, high ADIFF: measurement sensitivity is high and updates the a-priori estimate with new information about the target variable. This is by far the largest category defining ~79% of all retrievals on any given day of measurements. These retrievals are typically flagged "successful" in that they pass all CLIMCAPS quality control (QC) thresholds.

(2) High AK, low ADIFF: measurement sensitivity is high and agrees with the a-priori representation of the target variable. This is the second largest category (~17%) of CLIMCAPS retrievals.

(3) Low AK, low ADIFF: measurement sensitivity is low, so the retrieval predominantly resembles the a-priori estimate. This is the smallest category, defining only ~1% of retrievals on any given day.

(4) Low AK, high ADIFF: measurement sensitivity to the target variable is low and the spurious effects visible in ADIFF is predominantly error. The CLIMCAPS QC filters typically flag retrievals in this category as "failed.

When we evaluate retrieval continuity between CLIMCAPS-Aqua and CLIMCAPS-SNPP, we typically want to see similar space-time patterns in these metrics as it would mean that CLIMCAPS maintains consistency in retrieval SNR despite changes in instrumentation.



## 3 Experimental design

CLIMCAPS retrieves its large suite of atmospheric state parameters sequentially in a series of steps (Smith and Barnet, 2023a).
This step-wise approach holds many advantages, not least because it helps minimize the size of the a-priori error covariance matrix employed during retrieval for the sake of a computationally efficient algorithm that can support both real-time and full mission applications. Of primary concern to the CLIMCAPS V2 data record, however, is how such a sequential approach enables step-wise, targeted updates to the measurement error covariance matrix as knowledge about the true state and retrieval uncertainty grows (Smith and Barnet, 2019). The CLIMCAPS measurement error covariance matrix, $\mathbf{S}_m$, is the sum of the
radiance error covariance matrix, $\delta\mathbf{R}\delta\mathbf{R}^\mathrm{T}$, and radiance uncertainty due to the background atmospheric state, $\mathbf{K}^\mathrm{T}\mathbf{S}_b\mathbf{K}$, where $\mathbf{K}$ is the Jacobian and $\mathbf{S}_b$ the background state error covariance matrix (Smith and Barnet 2019, 2020). All atmospheric state parameters held constant during any given retrieval step, are considered "background" parameters. CLIMCAPS retrieves clouds first, followed by temperature [air_temp] and water vapor [h2o_vap] in that order (Smith and Barnet 2023a). Being first, the cloud properties are retrieved using only a-priori, not retrieved, estimates of the clear atmospheric state. Similarly,
the measurement error covariance matrix is, at first, defined only by a-priori estimates of radiance and background state uncertainty. After retrieval of the cloud parameters, the IR radiance measurements are cloud cleared (Smith and Barnet, 2023b) to allow sounding retrievals of the full, clear atmospheric column (i.e., characterization of atmospheric state past, not through, clouds). During this step (and each subsequent step), retrieval uncertainty is quantified, then propagated to the next step. Note that cloud clearing is the only step that updates the radiance uncertainty directly. In all subsequent steps, $\mathbf{S}_m$ will incorporate
the cloud-cleared radiance uncertainty $\delta\mathbf{R}_{cc}\delta\mathbf{R}_{cc}^\mathrm{T}$ instead of the a-priori measurement uncertainty $\delta\mathbf{R}\delta\mathbf{R}^\mathrm{T}$ (see Eq. 5 in Smith and Barnet, 2023b). Once [air_temp] is retrieved, $\mathbf{K}^\mathrm{T}\mathbf{S}_b\mathbf{K}$ is updated with the retrieved uncertainty of [air_temp]. More generally, a CLIMCAPS retrieval at step $x$+3 uses a background state with a-priori estimates replaced by the retrieved parameters from steps $x$, $x$+1 and $x$+2. In other words, the step-wise CLIMCAPS approach uses scene-specific signal *and* noise estimates as soon as they become available so that the a-priori state is gradually updated to represent retrieved conditions
within the FOR. The main purpose of this step-wise retrieval approach is to improve the efficiency with which the radiance measurement can be decomposed into discrete signals (Smith and Barnet, 2019).

The sequence and number of CLIMCAPS retrieval steps are not fixed but can be arranged to suit target applications. The CLIMCAPS V2 sequence broadly progresses as follows. (i) Cloud clearing is spatially linear, so cloud-cleared radiances are
retrieved first. (ii) [air_temp] is spectrally the most linear of all sounding parameters and it is important to have a stable, accurate representation of [air_temp] before attempting water and trace gas retrievals, so it is retrieved ahead of all trace gases. (iii) [h2o_vap] vapor is highly non-linear, but it can be retrieved with accuracy once [air_temp] is known. (iv) Ozone ($O_3$) is somewhat non-linear but has second order effects on temperature and water vapor, so it is retrieved next. (v) CO is highly linear with negligible impacts on other parameters. (vi) Nitric acid ($HNO_3$) is linear and impacts temperature in polar regions.
(vii) [air_temp] is then retrieved a second time from the same MERRA-2 a-priori to capture non-linearities due to retrieved





(not a-priori) knowledge of the value and error in [h2o_vap] and O₃. (viii) Finally, CO₂, N₂O and S₂O are retrieved in this sequence. Their spectral signals are mostly linear but the measurement information content for these gases is very low so we do not consider the retrievals of sufficient quality to affect the final [air_temp] retrieval. In future, we may change the order of these retrieval steps as we develop more accurate a-priori estimates.


Of primary importance in the CLIMCAPS product suite are the atmospheric profile retrievals from cloud-cleared radiances, so our experiment will start with an evaluation of cloud clearing uncertainty metrics that propagate and affect measurement SNR in all subsequent steps. Cloud clearing is a well-established method for addressing the way clouds affect IR radiation through the atmosphere (Chahine, 1977; Susskind et al., 1998) and is implemented in CLIMCAPS V2 as discussed in Smith

and Barnet (2023b). In short, cloud clearing aggregates the clear-sky radiance signal from each cluster of 3 x 3 FOVs (~15 km at nadir) to retrieve a single cloud-cleared radiance spectrum that represents the atmosphere within the larger FOR (~45 km at nadir). Figure 1 depicts the key uncertainty metrics CLIMCAPS generates during its cloud clearing step, namely etarej and ampl_eta. The former quantifies the bias in retrieved cloud-cleared radiance and the latter the degree to which random instrument noise is affected within the aggregated field-of-regard (FOR). If there are differences in the degree to which

instruments measure atmospheric clouds, we expect to see them amplified in these uncertainty metrics. The swath width of CLIMCAPS-SNPP (Figures 1a and 1c) is slightly wider than CLIMCAPS-Aqua (Figures 1b and 1d), hence the smaller data gaps at lower latitudes. Otherwise, we see that etarej and ampl_eta are largely consistent between CLIMCAPS-SNPP (Figures 1a and 1c) and CLIMCAPS-Aqua (Figures 1b and 1d) for a global day of data aggregated onto a 1° equal-angle grid. Note similarities in the range of values as well as the absence of any view angle effects (i.e., uncertainty does not increase with

view angle). Cloud clearing introduces systematic uncertainty (etarej > 0) and amplifies random instrument noise (ampl_eta > 1) wherever clouds are present. However, cloud clearing reduces random instrument noise (ampl_eta < 1) when clouds are absent and a cluster of measurement FOVs are simply averaged into a single FOR radiance spectrum (Smith and Barnet, 2023b). Figures 1c and 1d show that ampl_eta < 1 for much of the globe on any given day.

The differences that do exist between CLIMCAPS-SNPP and CLIMCAPS-Aqua with respect to ampl_eta and etarej can primarily be attributed to the time difference in observation between the two satellites, as clouds can change significantly over the course of minutes. In addition, there are small spatial shifts in how AIRS and CrIS observe clouds across their FORs since the AIRS and CrIS FOVs are not spatially co-registered. What we mean by this is that the 3 x 3 AIRS FOVs could observe a large radiance gradient compared to CrIS simply because of the sampling differences between the two instruments.

Moreover, the CrIS FOVs rotate with respect to each other as a function of view angle, while the AIRS FOVs maintain their aligned irrespective of view angle. This means that AIRS and CrIS do not necessarily observe the same cloud structure within each FOR. Even if the instruments are exactly the same (such as CrIS on SNPP and JPSS+), one can cannot compare their retrievals directly due to differences in instrument sampling, both spatial and temporal. This is true for all retrieval approaches,

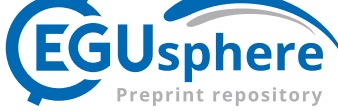

whether they employ cloud clearing or not. In CLIMCAPS, we only provide products in the overlap period between satellites

(see Table 2) for product evaluation where this aspect is properly accounted for. Overall, the large-scale similarities observed in Figure 1 do make sense because cloud clearing does not depend on spectral information content where most of the instrument differences manifest, or any a-priori knowledge of clouds that would amplify sampling differences. Instead, cloud clearing uses *spatial* information content (quantified as measurement variation across a cluster of FOVs) to derive an aggregate cloud-cleared spectrum (Smith and Barnet 2023b). AIRS and CrIS FOVs are ~14 km at nadir, and each instrument has 9 FOVs

making up a ~50 km cloud-cleared FOR. Moreover, both instruments have their FOVs arranged in 3 x 3 arrays, so they capture the available spatial information content in similar ways.

**Figure 1: CLIMCAPS V2 cloud clearing error metrics between (a+c) SNPP and (b+d) Aqua Level 2 retrievals on 30 August 2016.**
**These error metrics were binned, then averaged to a uniform 1° equal angle global grid. (Top row) [etarej] represents systematic error due to an incomplete removal of cloud signals from the Level-1B radiances, and (bottom row) [ampl_eta] quantifies the degree to which random instrument noise is amplified (or reduced) due to cloud clearing.**



For the purpose of this evaluation, there is no need to diagnose differences at finer spatial scales because our objective at this
stage is to identify (and address) the large-sale, systematic differences between CLIMCAPS-SNPP and CLIMCAPS-Aqua.
Addressing differences at finer scales will be the focus of future work.

Moving on to the next retrieval step, Figure 2 depicts CLIMCAPS temperature retrieval [air_temp] in the lower troposphere
for the SNPP and Aqua configurations.  CLIMCAPS employs the Stand-alone AIRS Radiative Transfer Algorithm (SARTA;
Strow et al., 2003) to simulate top-of-atmosphere radiances. SARTA was originally developed for AIRS, but later adapted for
CrIS. It is possible that SARTA introduces subtle effects into the retrieval product as a result of how it treats differences in
instrument spectral correlations. We should note that SARTA is not funded to be maintained consistently for AIRS and CrIS.
This means we can expect SARTA to introduce retrieval differences, which we will group under the umbrella of "instrument
effects" for the sake of simplicity and clarity of argument in this paper. Another instrument difference that may manifest in the
retrievals is the fact that AIRS requires a frequency correction – which is a function of orbital position and season. Figure 2
depicts no significant difference in the spatial pattern of temperature gradients, maxima and minima, between these two
CLIMCAPS configurations.

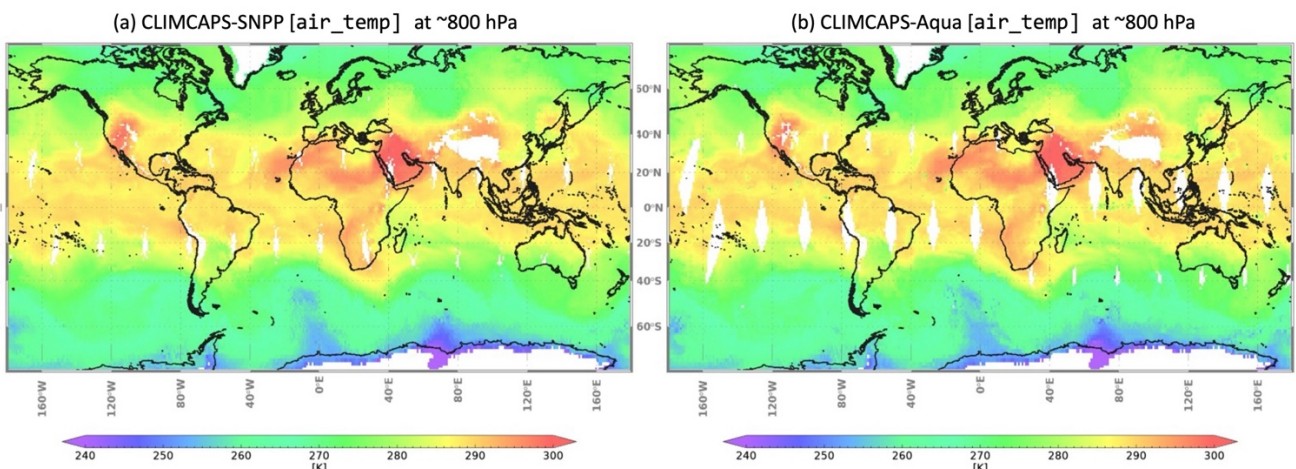

**Figure 2: CLIMCAPS V2 [air_temp] retrievals from (a) SNPP and (b) Aqua measurements acquired on 30 August 2016. The
CLIMCAPS [air_temp] retrievals are made on 100 pressure levels spanning the vertical atmospheric column from Earth surface
to top-of-atmosphere (~0.05 hPa). These figures represent all retrievals that passed quality control [ispare_2=0] and were made at
~800 hPa. For display purposes, the values were binned and averaged to a 1° equal angle global grid.**

The results in Figure 2 alone, however, do not satisfy the requirement for consistency nor does it demonstrate that we have
mitigated the differences between the Aqua and S-NPP systems sufficiently. CLIMCAPS employs dynamic regularization,
which means that it populates the true null space within each FOR with the a-priori estimate. A measurement with large null





space (or low information content and weak sensitivity to the target parameter) will yield a retrieval that approximates the a-priori estimate, a feature that generally applies to all OE retrieval systems. CLIMCAPS prepares its a-priori for temperature

by selecting the two MERRA2 reanalysis fields on either side of the measurement in space and time, and interpolating to the exact location. The CLIMCAPS V2 a-priori for temperature provides a good estimate of prevailing conditions even if no new information is contributed by the measurement. At this scale we, thus, expect the MERRA2 a-priori to neutralize any differences in observing capability between SNPP and Aqua. For an evaluation of retrieval consistency, we instead turn our attention to diagnosing the CLIMCAPS uncertainty metrics that quantify retrieval SNR where Aqua and SNPP instrument

would typically manifest. For this reason, all the results discussed in subsequent sections focus on an evaluation of CLIMCAPS V2 uncertainty, not its retrievals. This is a major departure from standard validation techniques that compare geophysical quantities from multiple, independent sources (e.g., Nalli et al., 2018; Wang et al., 2020) to determine their absolute accuracy. It is important to distinguish these two approaches and recognize their divergent end-goals. This said, the evaluation of uncertainty metrics presented in this paper is made possible largely *because* of the CLIMCAPS algorithm and product design.

Prior to 2020, we did not have access to 20+ years of AKs from the same system across different instrument suites; the NUCAPS operational product at NOAA still does not output its AK,s and AST-Aqua V7 is a single instrument product and optimized in isolation without any consideration for other instrument suites. Moreover, retrieval systems like NUCAPS and AST-Aqua V7 that retrieve their a-priori and final retrieval from the same measurement confound the analysis of the AKs. CLIMCAPS V2 AKs can be interpreted in a straightforward manner, and ADIFF is a simple arithmetic calculation.


Having established the target metrics for our evaluation in this paper, Figure 3 addresses questions regarding view angle and time-of-day, both of which potentially affect signal strength and thus continuity across instrument. We use CLIMCAPS-Aqua on 30 August 2016 as illustration but the same holds for CLIMCAPS-SNPP on the same day (not shown). The DOFS of a retrieval system with respect to a target variable is the sum-total of all corresponding AK peaks (or, the trace of the AK matrix).

When mapped out, as in Figures 3a and 3b, DOFS provides an efficient way of summarizing information content patterns. It is immediately obvious that the CLIMCAPS-Aqua DOFS for air_temp does not increase with view angle, nor does it change dramatically between the AM and PM orbits. We can attribute this consistency in CLIMCAPS DOFS across all view angles and time-of-day to the efficiency with which CLIMCAPS OE applies dynamic regularization. In a retrieval system without such ability, one may see DOFS increase with view angle because at high angles the measurements observe a larger portion

of the atmosphere and has more spectral channels sensitive to conditions in the mid- to lower troposphere. One can think of this as the channels traversing a thicker atmosphere at higher view angles due to a larger slant path, which causes the AK sensitivities to move higher up in the atmosphere. Channels sensitivity to the boundary layer at nadir may be sensitive to the mid-troposphere at edge-of-scan (50° for CrIS, 49.5° for AIRS). This means that the channels we select for profile retrievals need to have sensitivities that span the full vertical column at all view angles. At nadir, many of the channels may become

sensitive to surface conditions where CLIMCAPS error estimates are also high (due to larger errors in cloud, surface skin temperature and emissivity). For such channels, their information contribution may be minimal at nadir. At higher view angles,



however, their SNR improve due to degreasing errors in emissivity and surface skin temperature at higher altitudes (lower atmospheric pressure). There is similar variability in channel SNR with changes in atmospheric conditions (e.g., lapse rate) and seasonal cycles. The AST dynamic regularization approach we adopted in CLIMCAPS allows access to channel

information content whenever and wherever it is available. In other words, the AST dynamic regularization approach CLIMCAPS employs does not limit instrument spectral information based on a-priori assumptions or fixed parameterization. This is especially important for an observing system of the Planetary Boundary Layer (PBL), such as CLIMCAPS. The PBL affects human health and well-being first and foremost, so it is one of the primary considerations in CLIMCAPS. Over and above the issues we consider in this paper for V3 upgrades, the CLIMCAPS system allows ample opportunity for targeted

upgrades to its PBL observing capability in future.

The ADIFF for CLIMCAPS temperature at 700 hPa (Figures 3c and 3d) complements the DFS maps (Figures 3a and 3b) and illustrates that the degree to which CLIMCAPS adjusts the MERRA-2 a-priori with information from the measurements, is consistent across land and ocean, night and day, as well as nadir and edge-of-scan. Figure 3e depicts the vertical information

content for seven CLIMCAPS retrieval variables: temperature [air_temp], water vapor [h2o_vap], ozone ($O_3$), carbon monoxide (CO), methane ($CH_4$), carbon dioxide ($CO_2$) and nitric acid ($HNO_3$). These profiles represent the global average of the AK peaks (i.e., diagonal vector of AK matrix), with the error bars depicting the standard deviation of SNR across all types of conditions on a global day. For each of the variables represented in Figure 3e, the AK statistics are reported separately for the 1:30 am and 1:30 pm orbits to demonstrate how time-of-day does not impose significant differences. Therefore, we focus

our evaluation for the remainder of this paper on large ensembles of AKs that include all view angles and orbits to optimize continuity in the the CLIMCAPS V3 product for all scenes across multiple satellites.





**Figure 3: CLIMCAPS information content as a function of (a–d) view angle and (e) time-of day. (a) CLIMCAPS-Aqua degrees of freedom (DOFS) for [air_temp] from all FORs in descending orbits (13h30 LT) on 30 August 2016. (c) CLIMCAPS-Aqua [air_temp] difference between retrieval and a-priori (MERRA-2) at ~700 hPa for all descending orbits. (b+d) Same as (a+c) but for all ascending orbits (01h30 LT). CLIMCAPS DOFs are independent of view angle and time-of-day (e) CLIMCAPS-Aqua [air_temp] averaging kernel (AK) mean and standard deviation (error bars) from all (red) ascending FORs and (blue) descending FORs. The AKs represent the average (and standard deviation) of the diagonal vectors from the retrieval AK matrices. CLIMCAPS AK structure and variance are independent of time-of-day.**




There are CLIMCAPS algorithm components that we can test, and that we know influence retrieval SNR. Table 1 summarizes those that we identified as directly affecting the temperature retrieval, as well as two trace gases, $CO_2$ and $O_3$. Based on results presented in Smith and Barnet (2020), we know that the AKs from CLIMCAPS-Aqua and CLIMCAPS-SNPP have significant differences for these variables. Table 1 distinguishes six algorithm components (Column 1), and reports their values for the

two CLIMCAPS configuration as they are currently implemented for V2 as well as how we propose to change then for a future version V3+. The values reported in Columns "x1", "x2" and "x3" represent the experimental configurations we analyse and discuss in Section 4. In summary, the six algorithm components we test in this paper are as follows:

-    $B_{max}$ is employed during CLIMCAPS regularization (see Smith and Barnet, 2020 for details). In short, $B_{max}$ is an empirical term that informs the eigenvalue ($\lambda$) threshold as follows: $\lambda_c = {1}/{B_{max}^2}$ . During regularization,

CLIMCAPS applies SVD to the measurement SNR matrix, $\widetilde{K}^T S_m^{-1} \widetilde{K}$. All functions with $\lambda \geq \lambda_c$ are used in the retrieval without any regularization. Those functions with $0.05 < \lambda < \lambda_c$ are damped and all with $\lambda < 0.05$ are excluded from the retrieval. A larger $B_{max}$ imposes a lower threshold value ($\lambda_c$), which means a greater number of eigen functions pass into the retrieval without any regularization. $B_{max}$ should not be too large because all measurements have noise to filter out, nor should $B_{max}$ be too small because all successful measurements have

some information to contribute. Note that when we talk about $B_{max}$ in this paper we mean the $B_{max}$ value employed specifically for [`air_temp`] retrievals. CLIMCAPS associates a different $B_{max}$ for each retrieval parameter. Our focus in this paper is on [`air_temp`], so we will therefore not qualify each mention of $B_{max}$ in this paper.

-    [`RTAerr`] quantifies the radiative transfer algorithm (RTA) bias that propagates into CLIMCAPS retrievals, whenever a radiative transfer calculation is made. [`RTAerr`] applies to each channel individually and varies

according to the accuracy of transmittance and radiance calculation within the RTA. The [`RTAerr`] associated with a specific RTA will typically decrease as the RTA matures. Historically, the AST [`RTAerr`] is calculated offline using a large ensemble of data for specific instrument configurations. CLIMCAPS-SNPP uses the NOAA [`RTAerr`] calculated for NUCAPS-SNPP/JPSS+, whereas CLIMCAPS-Aqua uses the NASA [`RTAerr`] calculated for AIRS. CLIMCAPS-Aqua and CLIMCAPS-SNPP uses different versions of SARTA (due to funding limitations

that inhibits synchronized SARTA upgrades across all instrument configurations), so we do not expect the [`RTAerr`] to be universal for all CLIMCAPS configurations. We list the [`RTAerr`], averaged across all spectral channels, for CLIMCAPS-SNPP and CLIMCAPS-Aqua in Table 1 below. Note how the V2 [`RTAerr`] is on average an order of magnitude larger for CLIMCAPS-SNPP (~0.5 K) than CLIMCAPS-Aqua (~0.05 K). We investigate this disparity in Section 4 and demonstrate how a lower [`RTAerr`] for CLIMCAPS-SNPP improves

retrieval quality.




- [apodcor] defines the degree to which adjacent channels are correlated due to apodization of the CrIS interferograms (or, radiance measurements). AIRS is a grating spectrometer (Table A.1) and its measurements, therefore, are naturally apodized. CLIMCAPS applies Hamming apodization to CrIS measurements to impose localized spectral response function and thus remove a significant portion of the geophysical noise otherwise
present in the CrIS interferograms (Barnet et al., 2000, 2023)

- The last three rows summarize the number of channels within each spectral subset that CLIMCAPS uses in the retrieval of [air_temp], [CO₂] and [O₃] respectively. As with [RTAerr], CLIMCAPS V2 development benefited from existing efforts at NOAA and NASA to inform these channel selections. We critically evaluate these V2 channel sets here and make recommendations for future upgrades.


**Table 1: Summary of the CLIMCAPS algorithm configurations discussed in this paper. Each configuration is unique with respect to the six algorithm parameters depicted in Column 1. The "V2" configurations represent the CLIMCAPS V2 record available via NASA GES DISC (Table 2). The "x1" through "x3" configurations depict the experiments we performed and "V3+" is the**
**configuration we propose for the next CLIMCAPS release for AIRS+AMSU and CrIS+ATMS. Since CrIS+ATMS are identical on the SNPP and JPSS platforms, the SNPP-V3 configuration depicted here, also applies to JPSS-V3. The number of [air_temp] channels reported here represent those selected from the long-wave (LW) IR band only. As shown in Table S2, CLIMCAPS uses channels from all three bands for [air_temp] retrievals.**

| | CLIMCAPS-SNPP | | | | | CLIMCAPS-Aqua | | | |
|---|---|---|---|---|---|---|---|---|---|
| | V2 | x1 | x2 | x3 | V3+ | V2 | x1 | x2 | V3+ |
| [air_temp] SNR threshold [$B_{max}$]* | 0.2 | 0.8 | 0.175 | 0.175 | 0.175 | 0.25 | 0.8 | 0.15 | 0.15 |
| SARTA error spectrum [RTAerr]$^\Delta$ | ~0.5 K | ~0.5 K | 0 | 0 | 0 | ~0.05 K | ~0.05 K | ~0.05 K | ~0.05 K |
| Correlation factor for each set of 3 adjacent apodized channels [apodcor] | 1.0, 0.625, 0.133 | 1.0, 0.625, 0.133 | 1.0, 0.0, 0.0 | 1.0, 0.625, 0.133 | 1.0, 0.625, 0.133 | N/A | N/A | N/A | N/A |
| [air_temp] channels in LW band | 105 | 105 | 225 | 105 | 105 | 134 | 134 | 134 | 134 |
| Total [CO₂] channels$^\xi$ | 54 | 54 | 54 | 54 | 96 | 61 | 61 | 61 | 96 |
| Total [O₃] channels$^\xi$ | 77 | 77 | 77 | 77 | 77 | 40 | 40 | 40 | 73 |
| * (Smith and Barnet, 2020). $^\Delta$See Fig 4c and Table S1 for more details. $^\xi$See Figure 4 | | | | | | | | | |

Figures 4 and 5 provide graphic depictions of the algorithm elements summarized in Table 1. Figure 4a contrasts the V2 channel subsets for [air_temp] in CLIMCAPS-SNPP (red) and CLIMCAPS-Aqua (mustard). This is not the full channel set but rather just those channels from the 660–760 cm⁻¹ spectral range with sensitivity to temperature at pressure levels from Earth surface to top-of-atmosphere. It is immediately obvious that there are significant differences in the channels sets for or



`air_temp` between SNPP and Aqua. For the CLIMCAPS V2 implementation in 2019, we combined best practices at NOAA
for NUCAPS-SNPP and NASA for AST-Aqua V7, respectively. The differences in these channel subsets, therefore represent
their divergent approaches. The goal with all present and future upgrades to CLIMCAPS is to evolve the system to a place
where the retrievals represent the observing capability common to all instruments making up the full record. The blue lines in
Figure 4a represents the "x2" configuration we test for CLIMCAPS-SNPP. For a full list of the CLIMCAPS V2 channel sets
currently used for all retrieval parameters, see Table S2.


In addition to channel set differences that affect CLIMCAPS SNR, there is also the SARTA forward model error, [RTAerr],
typically calculated by the respective retrieval teams. Similar to the channel sets, [RTAerr] evolved separately for SNPP and
Aqua at NOAA and NASA, respectively. Figure 4c illustrates the NOAA [RTAerr] for SNPP that we implemented in
CLIMCAPS V2. Also, in Figure 4c is the [RTAerr] for Aqua, but this spectrum is invisible due to it being orders of magnitude
smaller the SNPP [RTAerr], and thus off scale in Figure 4c (see Table 1, Row 2). This tells us that [RTAerr] is a potential
source of discontinuity in CLIMCAPS V2 record and needs to be updated and normalized across both system configurations
for the sake of retrieval consistency across instrument suites.

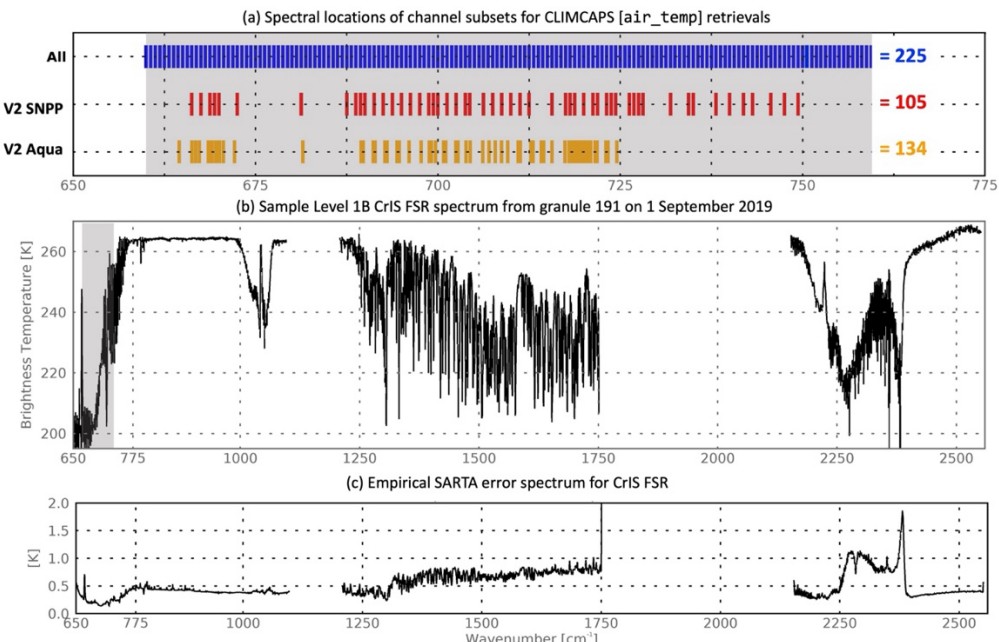

**Figure 4: (a) CLIMCAPS V2 spectral channel subsets used in [`air_temp`] retrievals for Aqua (orange) and SNPP (red) in the (grey area) 1.4 μm $CO_2$ absorption band. An experimental subset of channels used in the CLIMCAPS-SNPP x3 configuration (see 1) is in (blue). (b) Sample of CrIS full spectral resolution (FSR) measurement given by Level 1B product granule 191 on 1 September 2019 to contextualise the location of the 1.4 μm region in the CrIS long-wave band (~650–1100 cm⁻¹). (c) CLIMCAPS V2 empirically derived error spectrum of SARTA forward model bias for CrIS FSR spectra.**




Given known differences in CLIMCAPS V2 AKs for SNPP and Aqua (Smith and Barnet, 2020), we additionally diagnose channel subsets for $CO_2$ and $O_3$ in this paper. Figure 5 illustrates differences in the SNPP and Aqua V2 channel sets for $O_3$ and $CO_2$ (orange), and how we propose to standardize then for a future V3 implementation (blue).

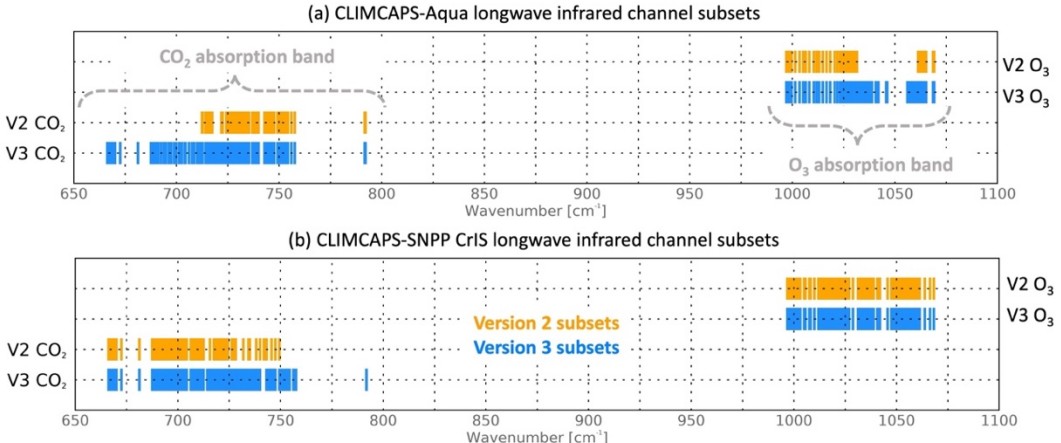


**Figure 5: Illustration of the channel subsets used for $CO_2$ and $O_3$ retrievals in the (orange) V2 and (blue) proposed V3 configurations for (a) CLIMCAPS-Aqua and (b) CLIMCAPS-SNPP (see Table 1).**

In this section we introduced and explained our experimental design because the evaluation of retrieval consistency across
different instrument suites is not straightforward. There is no universal retrieval approach that satisfies the SNR requirements of all applications, nor do we know exactly how instrument differences manifest under all conditions. The fact is that hyperspectral IR instruments measure a large array of atmospheric parameters that together characterize thermodynamic structure (i.e., temperature and water vapor profiles) and chemical composition (e.g., pollutant gas concentrations). But instead of a series of distinct spectral signatures, the result is a convolved IR measurement with complex inter-dependencies of both
geophysical signal *and* noise. CLIMCAPS mitigates these complexities to a large degree with its sequential retrieval approach and subsets of channels that maximize the SNR for each target parameter. But even then, the result is not clear-cut because instrument differences can cause spectral effects that are difficult to disassociate from the convolved geophysical signals. This means that one-to-one comparisons with independent datasets can be difficult to interpret. For a transparent evaluation of CLIMCAPS V2 retrieval consistency, we therefore focus on the analysis of CLIMCAPS uncertainty metrics instead. Table 1
summarizes all the algorithm components we tested, the results of which we present and discuss in Section 4 below.



## 4 Results and Discussion

In this section, we present results for each experimental configuration (Table 1) using available uncertainty metrics: AKs, DOFS and ADIFF. CLIMCAPS AKs typically have a large dynamic range across a day of retrievals because measurement SNR vary from scene-to-scene with prevailing conditions and CLIMCAPS regularization minimizes a-priori dependence

according to available information. Stated differently, when CLIMCAPS AKs have a small dynamic range for any given instrument configuration across a global day of measurements, we question whether the system is sufficiently optimized. We should note that most OE retrieval systems do not allow such a dynamic range in retrieval AKs because they constrain their solution to have a much smaller variation in their dependence on a-priori estimates, which can be large overall (e.g., Bowman et al., 2006; Irion et al., 2018).


Figure 6 depicts [air_temp] AKs for the existing V2 record as well as each of the experimental configurations listed in Table 1. When we talk about an AK, we mean the diagonal vector of a retrieval averaging kernel matrix that captures the peak AK values at each retrieval pressure level. The profiles in Figure 6 represent the average of all [air_temp] AKs on 1 September 2019 (321,300 in total), and the error bars are the standard deviation of the [air_temp] AKs. It is immediately

obvious in Figure 6a that there are systematic SNR differences between CLIMCAPS-SNPP (red) and CLIMCAPS-Aqua V2 (blue). Not only do the SNPP [air_temp] AKs lack vertical structure, they are also much lower in value as well as dynamic range compared to the Aqua [air_temp] AKs. This was first observed in Smith and Barnet (2020). The cause of this disparity is not immediately obvious and could be due to fundamental instrument differences in spectral resolution, over-damping of the measurement within retrieval system, or a combination of both.


We can diagnose the causal factors influencing V2 [air_temp] AK disparities by changing the retrieval configurations as listed in Table 1.  Figure 6b contrasts the SNPP V2 [air_temp] AK against three experimental configurations, x1, x2 and x3. When the SNPP $B_{max}$ threshold is increased from 0.2 to 0.8 (i.e., configuration x1) we see a dramatic jump in AK values across all pressure levels as well as a larger dynamic range in the mid-troposphere. As discussed in Section 3, $B_{max}$ is derived

empirically and informs the eigenvalue threshold ($\lambda_c$) that determines the number of eigen functions CLIMCAPS will damp (or leave undamped) in the retrieval step. $B_{max}$ is considered too large (and its corresponding $\lambda_c$ too small) when the AKs exhibit spurious effects and retrieval accuracy is low. This happens when CLIMCAPS uses too many eigen functions without any damping. An SVD of the measurement sensitivity matrix, $\widetilde{\mathbf{K}}^{\mathrm{T}}\mathbf{S}_m^{-1}\widetilde{\mathbf{K}}$, results in eigen functions arranged in order of signal strength such that higher order eigen functions are associated with signal and lower order functions are dominated by noise.

When lower order eigen functions are not sufficiently damped, then their noise propagates into the retrievals as error. Similarly, a $B_{max}$ value is considered too small (and $\lambda_c$ too large) when too many higher order eigen functions are damped and not enough measurement information passes on to the retrieval. In such a case, the AKs will be small with a low dynamic range and retrievals will mostly represent the a-priori. An evaluation of $B_{max}$, therefore, needs to include an analysis of retrieval accuracy.



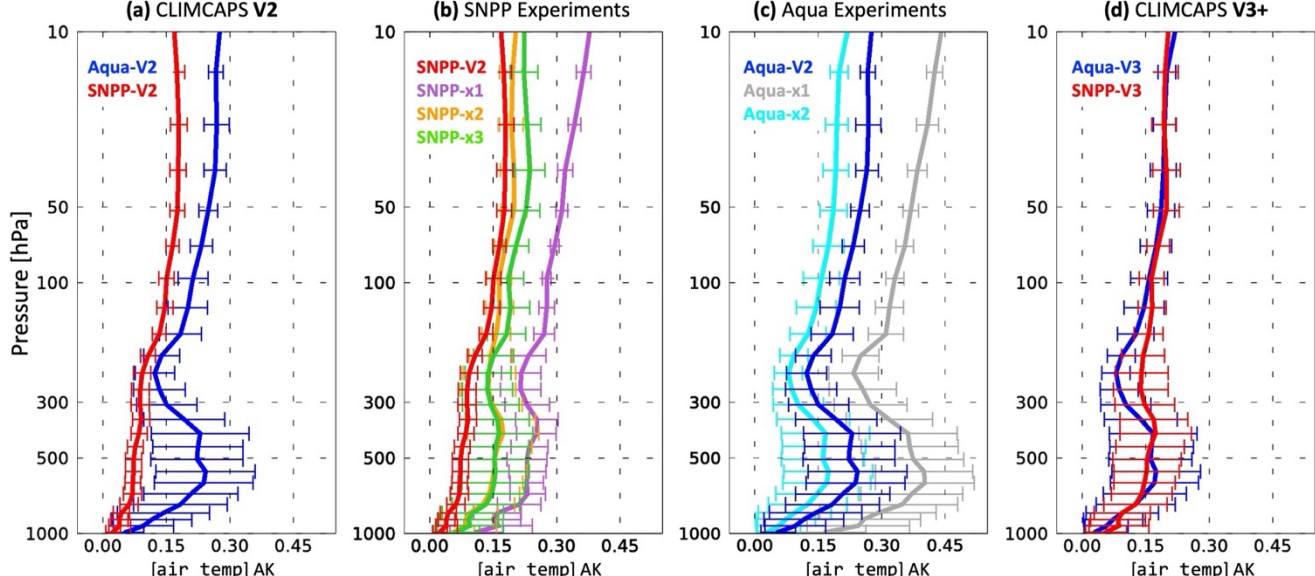


**Figure 6: Averaging kernel (AK) comparisons to illustrate disparities in [`air_temp`] information content structure and variance for (a) CLIMCAPS V2 (blue) Aqua and (red) SNPP, (b) three CLIMCAPS-SNPP experimental configurations (purple+orange+green) contrasted against the (red) V2 SNPP configuration, (c) two CLIMCAPS-Aqua experimental configurations (cyan+grey) against the (blue) V2 Aqua configuration, and (d) the proposed CLIMCAPS V3 configurations (blue) Aqua and (red)**
**SNPP to improve Level 2 data continuity. (b+c) The experimental configurations referenced here are detailed in Table 1.**

What is evident in Figure 6b is that $B_{max}$ alone does not explain the low AK values for CLIMCAPS-SNPP V2 (red). Both the x2 and x3 configurations for CLIMCAPS-SNPP (green and orange) have slightly lower $B_{max}$ values (0.175), yet higher AKs with larger variance. This is the oppositive of what we expect for smaller $B_{max}$ values (see earlier discussions). Using the

SNPP-V2 configuration ($B_{max}$= 2.0) and changing only the [`RTAerr`] to 0 K, we see a significant change in the AK structure, magnitude and variance (not shown). This tells us that the NOAA [`RTAerr`] we adopted for CLIMCAPS-SNPP is much too large (Figure 4c). In fact, it is an order of magnitude larger, on average, than the [`RTAerr`] for CLIMCAPS-Aqua (~0.05 K). We, therefore, propose that the [`RTAerr`] should be carefully rerecomputed for all CLIMCAPS V3+ CrIS+ATMS configurations. For the sake of this paper, however, we list [`RTAerr`] = 0 as the placeholder value for future upgrades, since

the effort required to achieve an accurate estimate of [`RTAerr`] is out of scope of this paper.

Another question that often arises with regard to CLIMCAPS-SNPP/JPSS+ is the issue of apodization. We will not explain apodization in this paper, nor the reasons for apodizing CrIS radiances ahead of retrievals. Others cover this great detail (Barnet et al., 2000, 2023). What we do wish to illustrate here is how spectral correlation due to apodization has no significant impact

on CLIMCAPS AKs and, therefore, retrieval SNR. The SNPP-x2 and SNPP-x3 configurations differ only in the number of



long-wave (LW) IR channels used in [air_temp] retrievals. SNPP-x2 uses all available CrIS channels in the 660–760 cm⁻¹ range (225 channels in total) while SNPP-x3 uses the V2 105 channel subset (Figure 4a). In addition, SNPP-x2 sets [apodcor] = 0 for all adjacent channels to indicate the absence of spectral correlation due to apodization. A comparison of the AKs (Figure 6b) and [air_temp] root-mean-square-error (RMSE, Figure 7a) show no significant difference in retrieval results between SNPP-x2 and SNPP-x3. We, therefore, do not recommend any changes to the apodization of CrIS radiances, nor the V2 [air_temp] channel subsets for future CLIMCAPS upgrades. The SNPP-x1 and Aqua-x1 configurations yielded very high AK values, but because their B_max values forced CLIMCAPS to use a large number of eigen functions undamped, their retrieval quality deteriorated to the point where their [air_temp] RMSE values exceeded the range depicted in Figure 7 (not shown).

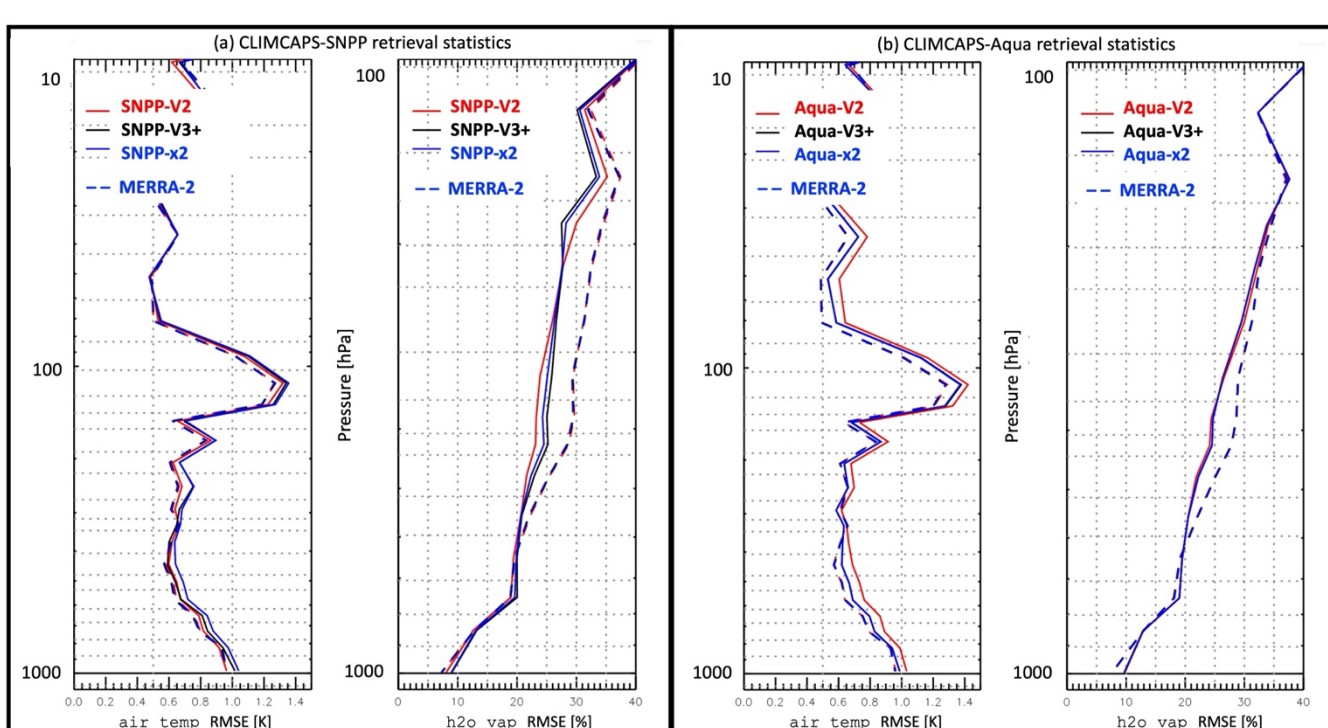

**Figure 7: Statistical analysis of [a] CLIMCAPS-SNPP and [b] CLIMCAPS-Aqua atmospheric temperature [air_temp] and water vapor h2o_vap retrievals. The profiles represent the root-mean-square error (RMSE) of all retrieved values (321,300 in total) against the corresponding reanalysis fields from GFS (Global Forecast System) on 1 September 2016. Each profile represents a different configuration; [solid red] V2, [solid black] V3+ and [solid blue] x2 for each respective instrument configuration as outlined in Table 1. The striped lines are the RMSE of MERRA-2 against GFS. CLIMCAPS interpolates MERRA-2 in space and time to each instrument grid before using it as a-priori estimate for [air_temp] and [h2o_vap]. The RMSE of [air_temp] is reported in units Kelvin [K] for the troposphere and lower stratosphere (1000–10 hPa) and the RMSE of h2o_vap is reported as percentage error [%] for the troposphere only (1000–100 hPa).**





Figure 7 depicts the RMSE of all successful CLIMCAPS [air_temp] and [h2o_vap] retrievals made on 1 September 2016 against the National Center for Environmental Prediction (NCEP) Global Forecast Model (GFS) (Wang et al., 2019). For CLIMCAPS-Aqua, the RMSE values are largely unchanged in the [h2o_vap] retrievals across all three configurations depicted in Figure 7b; V2, x2 and V3+. The RMSE for CLIMCAPS-Aqua [air_temp] retrievals are exactly the same between x2 and V3+, but V2 deviates markedly across most of the vertical atmospheric column. CLIMCAPS-Aqua V3+ [air_temp] has higher accuracy than CLIMCAPS-Aqua V2 [air_temp]. This is interesting because the V3+ $B_{max}$ is lower (0.15) than what is currently used in V2 (0.25). What these results could suggest is that CLIMCAPS-Aqua V2 [air_temp] retrievals are under-damped with too many lower order eigen functions contributing their noise to the retrieval. A lower $B_{max}$ value results in a higher eigenvalue threshold ($\lambda_c$), which means a smaller subset of higher order eigen functions contributing their information to the retrieval. Alternatively, these results for CLIMCAPS-Aqua [air_temp] could suggest that the V2 retrievals ($B_{max} = 0.25$) are more accurate in its deviation from the MERRA-2 a-priori, but that the GFS and MERRA2 temperature fields have a strong agreement, such that the retrieval RMSE is low whenever [air_temp] ADIFF is low. We recognize that it is an onerous task to optimize $B_{max}$ in absolute terms due to the absence of a reference dataset depicting true atmospheric conditions at the time of satellite overpass and within the instrument footprint. The best we can do is adopt an empirical approach for determining $B_{max}$ in relative terms by comparing retrieval results to a high quality, independent global dataset like GFS. Note that the values we suggest here for consideration in CLIMCAPS V3 would need to be verified across a larger sample of focus days before implementation.

In general, our philosophy for CLIMCAPS [air_temp] is to maintain a strong dependence on MERRA-2, unless the measurement SNR is very high. We adopted this philosophy in CLIMCAPS V2 to ensure a stable long-term thermodynamic baseline for the suite of CLIMCAPS trace gas retrievals; [h2o_vap], $O_3$, $CO_2$, $CO$, $CH_4$, $N_2O$, $HNO_3$ and $SO_2$. This is because [air_temp] influences all subsequent retrievals for any thermal sounder (Smith and Barnet, 2019, 2023a). This is demonstrated for [h2o_vap] in Figure 7b, where the small variations in [h2o_vap] RMSE across all three configurations can only be attributed to corresponding changes in [air_temp] since we did not vary any algorithm components for [h2o_vap]. As discussed elsewhere (Smith and Barnet, 2020, 2023a), CLIMCAPS retrieves [h2o_vap] after [air_temp] to lower the associated background error estimate for [air_temp] in $\mathbf{S}_m$ to improve [h2o_vap] SNR. The results for [h2o_vap] in Figures 7a and 7b reflect lower RMSE values in the mid- to upper troposphere for both CLIMCAPS-Aqua and CLIMCAPS-SNPP , but show larger variation across the three configurations of CLIMCAPS-SNPP (Figure 7a, righthand panel). This can be attributed to changes in both [air_temp] $B_{max}$ and [RTAERR].

We can turn our attention to evaluating ADIFF next (Figure 8). As discussed, ADIFF quantifies the difference between retrieval and a-priori. When we look at a global map of ADIFF at a specific pressure level, we can gain insight into the retrieval system. For example, an ADIFF with a consistent speckle pattern across large areas would indicate random retrieval SNR, or an ADIFF





with strong latitudinal pattern would indicate systematic bias in retrieval SNR. What we want to see in a global map of ADIFF,
instead, is consistency across latitudes with a spatial pattern corresponding to known geophysical features, like clouds.

**Figure 8: CLIMCAPS [`air_temp`] difference between retrieval and a-priori (MERRA-2) at ~700 hPa on 30 August 2016 for all ascending (13h30 LT) and descending (01h30 LT) orbits aggregated to a uniform 1° equal-angle global grid. Comparison of**
**[`air_temp`] differences, ADIFF, between (a+b) CLIMCAPS V2 and the proposed (c+d) CLIMCAPS V3 configurations for (a+c) SNPP and (b+d) Aqua.**

Figures 8a and 8b depict V2 [`air_temp`] ADIFF at 700 hPa for the SNPP and Aqua configurations, while Figures 8c and 8d
show what ADIFF would look like for a V3 system. It is immediately obvious the CLIMCAPS retrieval SNR is neither truly
random nor alarmingly biased. This said, the V2 [`air_temp`] ADIFF has large differences between the two instrument
systems, both in magnitude and dynamic range. This indicates that there are systematic differences in the way CLIMCAPS V2
is configured for CLIMCAPS-SNPP and CLIMCAPS-Aqua, but that these are largely resolved in the proposed V3



[`air_temp`] configuration. A stronger correlation between CLIMCAPS-SNPP and CLIMCAPS-Aqua in iV3 [`air_temp`]
ADIFF tells us that the upgrades we suggest here, will improve continuity in the CLIMCAPS record across CrIS+ATMS and
AIRS+AMSU. Even so, we do not expect to fully resolve retrieval continuity across all parameters from CrIS+ATMS and
AIRS+AMSU, given the list of known differences in instrumentation (Table S1). Some of the other changes we suggest for a
CLIMCAPS V3 upgrade (see Table 1) pertain to the channel subsets for $O_3$ and $CO_2$ retrievals, which we illustrate in Figure
9 below.

**Figure 9: CLIMCAPS AKs for [`air_temp`] as well as six atmospheric gases, [`h2o_vap`], $O_3$, CO, $CH_4$, $CO_2$ and $HNO_3$, as the mean**
**and standard deviation (error bars) for all retrievals in ascending and descending orbits that passed quality control on**
**1 September 2016. A comparison of CLIMCAPS AKs from (blue) Aqua and (red) SNPP using three different configurations, (a) V2,**
**(b) experimental SNPP-x3 and Aqua-x2, as well as the proposed (c) V3 for both instrument suites.**





Figure 9 depicts the global average (and standard deviation) of CLIMCAPS AKs for seven retrieval parameters –
[air_temp], [h2o_vap], $O_3$, CO, $CH_4$, $CO_2$ and $HNO_3$ – for the different configurations evaluated in this paper (Table 1).
Figure 9a represents the V2 retrieval capability we discussed in previous work (Smith and Barnet, 2020). Apart from
[air_temp] and [h2o_vap], we also notice disparities in CLIMCAPS V2 AKs for $CO_2$ and $HNO_3$. The disparities that exist
for the CO AKs across CLIMCAPS-Aqua and CLIMCAPS-SNPP, on the other hand, are predominantly due to known IR

instrument differences; the AIRS shortwave (SW) IR band does not cover the full spectral absorption region for CO, while
CrIS captures all spectral channels sensitive to CO (Table 1, Smith and Barnet, 2019). There is, thus, a physical limit to the
signal that exists for CO in the AIRS sounder. We argue that it is important to capture as much of the CO signal as possible,
given the present-day need for information about air quality and fire emissions. We, therefore, propose to make no algorithm
changes to CLIMCAPS-SNPP/JPSS+ that would reduce the CO AK for the sake of mimicking CLIMCAPS-Aqua CO

capability. This is the only retrieval parameter for which we make this exception. All other retrieval parameters are supported
by comparable instrument observing capabilities.

Figure 9b illustrates AK differences for CLIMCAPS-SNPP x3 and CLIMCAPS-Aqua x2 (see Table 1 for details). With the
[RTAERR] reduced to zero from ~0.5 K on average in V2, we see changes manifest across all retrieval parameters. This is

due to the fact that the CLIMCAPS [RTAERR] is a spectrum (Figure 4c) that affects all channels. What is especially
encouraging is the fact that the $HNO_3$ AK differences visible in V2 (Figure 9a) is mostly resolved as a result (Figures 9b and
9c). Changes to CLIMCAPS-SNPP [RTAERR] also has dramatic effects on the $O_3$ and $CO_2$ AKs. This meant revisiting the
channel subsets for these two gaseous species for both CLIMCAPS-SNPP and CLIMCAPS-Aqua to promote consistency in
observing capability. The $O_3$ and $CO_2$ channel subsets we propose for V3 (Figure 5), has a marked effect on retrieval AKs and

indicates that the differences that do exist can be mostly resolved as illustrated in Figure 9c.

**5 Conclusions**

With this paper we demonstrated how decadal continuity can be achieved in a Level 2 product using the same retrieval system
and measurements from two sounding satellites with different technologies, AIRS and CrIS. Achieving retrieval consistency
across instruments is not a trivial task and requires knowledge of instrument design as well as calibration. Like all atmospheric

sounding observations, CLIMCAPS retrievals are not direct measurements but, instead, indirect observations derived from
inverting the top-of-atmosphere IR and MW spectra. Many signal inversion techniques exist depending on the type of
measurement and target application. For CLIMCAPS, we employed the method originally developed by Susskind et al. (2003)
within the AST for the AIRS instrument. We emphasized how the AST approach to dynamic, scene-dependent OE
regularization, benefits CLIMCAPS retrievals and contrasts with the static, generalized regularization promoted in Rodgers

(2000). CLIMCAPS uses a space-time collocated reanalysis model, MERRA-2 (Gelaro et al., 2017), as a-priori estimate for
[air_temp], [h2o_vap] and $O_3$. Not only does this provide a-priori estimates that are largely independent of the





measurements, it also allows the calculation of representative a-priori error covariance matrices for use in the physical OE retrieval steps. In contrast, the NUCAPS and AIRS V7 systems use statistical operators to generate a-priori estimates directly from the instrument measurements. NUCAPS employs a linear regression (Goldberg et al., 2003) and AIRS V7 a neural
network retrieval (Milstein and Blackwell, 2016). Statistical a-priori estimates that are derived from the spectral measurements, confound the analysis of AKs since instrument information is contained both in the a-priori estimate and final retrieval. One could argue that reanalysis models, like MERRA-2, already assimilate spectral information from AIRS, CrIS, AMSU and ATMS to derive their model fields. But, as argued in Smith and Barnet (2019), not only do reanalysis models, like MERRA-2, assimilate only a small subsets of spectral channels, they also apply rigorous spatial thinning to avoid any interference from
clouds, aerosols or smoke. In addition, a MERRA-2 assimilates spectral information from a large array of sources so the contribution from any individual instrument at a specific space-time location is always low, if now absent. The CLIMCAPS design, with its instrument-independent a-priori and representative error covariance matrices, enables the systematically propagating of error through all retrieval steps to yield accurate retrievals with high yield of successful results across the globe (Smith and Barnet, 2019).


The Level-2 CLIMCAPS V2 product contains the AK matrices for each retrieved parameter, at each instrument FOR, which enables the SNR analysis. The validation of retrieved quantities alone does not advance knowledge of a retrieval system. Instead, the analysis of SNR we presented in this paper offers a more honest, in-depth assessment of the contribution IR measurements make to the retrieved solution. This, in turn, helps promote retrieval consistency since the results can be used
to optimize CLIMCAPS to have consistency in measurement contribution (as characterized by the shape, magnitude and variance of the AKs) across a wide range of environmental conditions and despite instrument differences.

We proposed a series of changes that can be implemented in CLIMCAPS V3, should future funding allow, to improve its multi-decadal sounding record. But over and above these specific recommendations, our paper is also relevant to other
sounding systems and instruments, such as the NOAA NUCAPS systems. If NOAA optimizes NUCAPS-MetOp and NUCAPS-JPSS+ for consistency across instruments, then their soundings can depict diurnal changes and help inform severe weather forecasts. Moreover, the modern-era approach is to de-aggregate large multi-sensor satellites into smaller IR-only and MW-only satellites, and to merge measurements from low-Earth orbiting and geostationary satellites alike. This means that there is a greater need for the ability to algorithmically handle and evaluate these new dynamic types of "representation" error
in retrievals. Also, it is expected that future instruments will have new kinds of instrument errors due to the higher demands on spatial and spectral resolution. The ability to robustly and efficiently account for instrument errors along with the ability to diagnose their impact on the retrieval products is the future of sounding science. The CLIMCAPS system and analysis approach we presented here is an initial step along this path.



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

**Competing interests**

The contact author has declared that none of the authors has any competing interests.

**Acknowledgements**

The work reported in this paper requires dedicated expertise and effective teamwork within a collaborative environment. The authors, Drs. Nadia Smith (NS) and Chris D. Barnet (CDB), therefore wish to express their sincere gratitude to NASA for funding the science and development of CLIMCAPS. Each milestone was met within the confines of the NASA ROSES (Research Opportunities in Space and Earth Science) competitive grant system spanning nine years altogether. Each grant
funded at most 1 FTE per year for a total of three years.

  2015–2018 (Principal Investigator CDB): ROSES grant NNH15CM66C enabled development of the retrieval code as well as the set of a-priori estimates.

  2018–2021 (Principal Investigator CDB): ROSES grant 80NSSC18K0975 enabled the set-up and operationalization of CLIMCAPS for instruments on Aqua, SNPP and JPSS.

2021–2024 (Principal Investigator NS): ROSES grant 80NSSC21K1959 supported the maintenance of CLIMCAPS, which included engagement with stakeholders to improve product design and develop applications.



The authors would like to acknowledge Joel Susskind who transformed techniques used in numerical quantum mechanics to provide the adaptive method of regularization that enables the robust utilization of hyperspectral satellite soundings employed in the operational AST, NUCAPS and CLIMCAPS algorithms. This approach continues to be refined and is still evolving, as demonstrated in this paper. The authors would also like to acknowledge some of the early influential work by Dr. Barney Conrath (e.g., Conrath, 1977) to understand the physical nature of information content. CDB had the honor to work closely with both Joel and Barney for many years and values their desire to maximize the exploitation of space sounding assets.

**Data Access**

Table 2 summarizes the CLIMCAPS V2 product suite that is maintained, archived and distributed by the NASA Goddard Earth Sciences Data and Information Services Centre (GES DISC). CLIMCAPS retrieves atmospheric soundings from AIRS+AMSU on Aqua as well as CrIS+ATMS on SNPP and the JPSS series to form a continuous, long-term record of atmospheric state variables (Table 2, column 3). In addition, CLIMCAPS generates two experimental products that span the full lifetimes of the two early satellites, namely (i) Aqua by predominantly using AIRS channels to overcome the loss of key AMSU channels in 2016 (Table 2, row 3), as well as (ii) SNPP by using CrIS normal (or low) spectral resolution (NSR) radiances (Table 2, row 4). Between 20 January 2012 and to 2 November 2015, the SNPP CrIS measurements were exclusively available in NSR mode, whereafter, a second, "full spectral resolution" (FSR) CrIS Level 1B product was added to better match the spectral information content of AIRS. The CLIMCAPS V2 record includes these two experimental datasets to support sounding science and instrument investigations but will not be released or maintained in future CLIMCAPS versions.

**Table 2: Summary of the full CLIMCAPS V2 record that is publicly available via the NASA data and information service centre at the Goddard Spaceflight Centre (or GES DISC). The shaded areas denote experimental products for sounder science and instrument design.**

| CLIMCAPS | GES DISC record length | CLIMCAPS L2 continuity product | CLIMCAPS L2 datasets [1]GES DISC shortname (DOI) | Input [2]L1B datasets GES DISC shortname (DOI) |
|---|---|---|---|---|
| **Aqua** AIRS+AMSU | 2002/08/31– 2016/09/25 | **2002/09/01— 2016/08/31** | SNDRAQIML2CCPRET (10.5067/JZMYK5SMYM86) | AIRS: AIRIBRAD (10.5067/YZEXEVN4JGGJ) AMSU: AIRABRAD (10.5067/LFUQ1L3IYVQD) |
| **Aqua** AIRS-only | 2002/08/31– present | N/A | SNDRAQIL2CCPRET (10.5067/ILFPVBTDHTDL) | AIRS: AIRIBRAD (10.5067/YZEXEVN4JGGJ) |
| **[3]SNPP** CrIS+ATMS NSR | 2012/01/20– 2021/05/21 | N/A | SNDRSNIML2CCPRETN (10.5067/9HR0XHCH3IGS) | CrIS NSR: SNPPCrISL1BNSR (10.5067/N9J1D8VZVJUX) ATMS: SNPPATMSL1B (10.5067/HFDD6A30MA10) |




| | | | | |
|---|---|---|---|---|
| **[4]SNPP** CrIS+ATMS FSR | 2015/11/02– 2021/05/21 | **2016/09/01— 2018/01/3** | SNDRSNIML2CCPRET (10.5067/62SPJFQW5Q9B) | CrIS FSR: SNPPCrISL1B (10.5067/9NPOTPIPLMAW) ATMS: SNPPATMSL1B (10.5067/HFDD6A30MA10) |
| **[5]JPSS-1** CrIS+ATMS | 2018/02/17– present | **2018/02/01— present** | SNDRJ1IML2CCPRET (10.5067/LESQUBLWS18H) | CrIS: SNDRJ1CrISL1B (10.5067/EETSCFBDBLX6) ATMS: SNDRJ1ATMSL1B (10.5067/VP66V3OTXOPY) |

[1] NASA Goddard Earth Sciences Data and Information Services Center, https://disc.gsfc.nasa.gov/.

[2] Level 1B calibrated, geolocated radiance measurements

[3] Suomi National Polar orbiting Partnership (SNPP) Normal Spectral Resolution CrIS mode

[4] SNPP Full Spectral Resolution (FSR) CrIS mode introduced to allow science-quality CO retrievals (Gambacorta et al., 2014)

[5] The first of four Joint Polar Satellite System (JPSS) payloads in space. In operational mode, JPSS-1 is known as NOAA-20