# Peer review of "An information content approach to diagnosing and improving CLIMCAPS retrieval consistency across instruments and satellites"

_EGUsphere, 2024_

## Referee Comment (RC2)

**The paper introduces new settings related to the CLIMCAPS retrieval for different instruments and satellites. It is well-structured and provides detailed explanations, but some sections are overly lengthy and could be revised for conciseness. Additionally, there are several minor corrections that need to be addressed.**

**major revisions:**

- The abstract provides extensive details about the instruments, satellites, and the CLIMCAPS system, but it lacks a focus on the results and the novelty of the paper. The results and their implications are only briefly summarized. Please conside a more balanced abstract.

- The introduction is too long, spanning about three pages. It contains excessive background information, such as discussions on MODIS, VIIRS, and CERES, which are not used in this paper and are therefore irrelevant. The introduction could be condensed to focus more on the study's specific goals and context.

- Section 2 is also too lengthy. For example, there is a detailed explanation of NUCAPS and AST-Aqua V7, which, while useful for comparison, could be summarized more. The discussion of dynamic regularization and SVD could also be more concise.

- Section 3, Experimental Design, also repeats certain concepts multiple times. For instance, the explanations about the sequential retrieval process and the covariance matrix are repeated unnecessarily.

**minor correction:**

- Line 63: NASA AST: what is AST stand for? It was not mentioned in the paper.
- Line 117: What is NUCAPS stand for?
- Line 184: The acronym AKs is introduced for the first time in line 184, but its meaning is only explained later in Section 2.3. and again later in Fig.6 (section 4) was mentioned about Averaging Kernel. Please consider this inconsitency.
- Line 274: The same story for „FOR", it was defined for the 1st time in line 274 but it was used several time before it for example in line 210 , …

---

## Author Response (AR2)

**Author response to Reviewer 1**

[Reviewer 1]: This manuscript describes use of several metrics (cloud clearing, degrees of freedom (DOF) from the averaging kernel matrix, and the amplitude of the retrieval update related to the prior) to evaluate thermal IR retrievals. These metrics are applied to output from the operational V2 CLIMCAPS retrieval system applied to two sensor systems (AIRS/Aqua and CrIS/SNPP) along with several experimental versions of CLIMCAPS. Comparisons between output from the different experimental versions leads to an explanation for some of the inconsistency noted in CLIMCAPS output from the two sensors, noted in earlier work (Smith and Barnet 2020). The result is a plan for a potential version 3 CLIMCAPS that would have improved consistency between output from the two sensors.

[Reviewer 1]: The paper is generally well structured and written and appropriate for AMT. I suggest some minor revisions to improve the clarity and presentation, particularly for the figures. Minor grammar corrections and some suggested rewordings are listed at the end.

[Response]: We would like to thank the reviewer for their review of our work. We value their astute observations as well as their attention to detail. Below we present our response to each point raised.

Minor revisions
* * *
[Reviewer 1]: The aim of the analysis of the experimental CLIMCAPS versions was to improve the *consistency* between the AIRS and CrIS results; the results are in important step forward in that regard, and I think the manuscript title should more directly reflect that. If possible at this stage, I would suggest changing the manuscript title to: "An information content approach to diagnosing and improving consistency of CLIMCAPS retrievals across instruments and satellites".

[Response]: The title now reads: "An information content approach to diagnosing and improving CLIMCAPS retrieval consistency across instruments and satellites"

[Reviewer 1]: My only technical critique are the two paragraphs in section 4 that deal with Figure 7 (lines 566 - 594). Figure 7 contains RMSE relative to GFS; without substantial other analysis comparing GFS and MERRA-2, it is very unclear what these plots are revealing. If a lower RMSE relative to GFS is evidence of "higher accuracy" of CLIMCAPS, that must be making an implicit assumption that GFS is itself more accurate than MERRA-2. I am not an expert on NWP models, but that assumption needs extra support and references. If GFS and MERRA-2 have similar accuracy, or if MERRA-2 has better accuracy, then the RMSE metric relative to GFS would not tell us anything about CLIMCAPS. I feel this section is not adding anything to the paper in the current form. My suggestion is to remove this figure and the two corresponding paragraphs from the paper; otherwise this needs more explanation to describe what this is revealing about the CLIMCAPS retrievals.

[Response]: Removed.

[Reviewer 1]: At the beginning of section 2 (Lines 116 - 124) there is description of some challenges of developing the CLIMCAPS retrieval system. This seems out of place here, and would be better included as part of the discussion at the end of the paper. Also, one of the unique aspects of CLIMCAPS is that effectively spans across two missions - this introduces some funding/support challenges as it seems NASA support is generally mission-centric. It may be useful to expand on this issue since the authors have first hand experience here.

[Response]: In recent years NASA funding sources have evolved away from mission-oriented algorithms and moved towards application or science-oriented algorithms. CLIMCAPS was originally funded as a continuity product designed for S-NPP and JPSS/CrIS+ATMS space-borne assets utilizing NASA's large and unique investment in sounding science culminating with a diverse set of Aqua/AIRS+AMSU products. The idea of a CLIMCAPS "facility algorithm" was to incorporate the sounding science expertise of the NASA Sounder Science Team's diverse set of user products within a multi-satellite product system.

PIs of science-oriented algorithms are usually experts on the specific science application and are not likely to be experts on sounding science or possess knowledge of subtle instrument characteristics or operational processing requirements. This can lead to inefficient and/or fragmented algorithms. It is both difficult and expensive to integrate a suite of disparate algorithm types given the different schedules, expertise, and limited funds within each science application.

On the other hand, algorithm developers are not necessarily fully aware of the subtle needs of a specific science application. With the evolution of NASA ROSES funding we attempted to be PI's and co-I's on as many science-oriented proposals as we could; however, given limited funds none of those were funded. We also agreed to be collaborators on numerous science proposals; however, these collaborations tend to be superfluous to sounding science and, as it turns out, unsustainable. We therefore, encapsulated our awareness of science applications with the core concepts of sounding science within a maintenance proposal - which ended up our only source of funding for the past three years. CLIMCAPS has been extremely robust and we did not have to make many changes related to instrument issues, instrument degradation, or algorithm bugs. Therefore, we used our maintenance funding to collaborate and provide upgrades as improvements to as many product applications as we could - including a major re-organization of the level-2 and level-3 product files. The collaborative efforts are discussed in numerous publications, some cited herein. This paper discusses those upgrades relevant to sounding science itself and how improvements in one product can translate into the other products in the multi-dimensional sounding space.

[Reviewer 1]: At line 372-375 there is a short section about the Planetary Boundary Layer. None of the analysis in the paper is targeted at PBL performance, so this is distracting from the central themes of the paper. I would recommend removing these few sentences.

[Response]: We removed these sentences from this section, and reworked it into the Conclusion since the work described here is relevant to PBL data product design.

[Reviewer 1]: Figures 1,2,3,8 are using Rainbow colormaps, which do not faithfully represent the data and are not interpretable by readers with color vision impairment: https://hess.copernicus.org/articles/25/4549/2021/, https://doi.org/10.1175/BAMS-D-13-00155.1, https://matplotlib.org/stable/users/explain/colors/colormaps.html
Instead, use a perceptually uniform colormap, or a diverging colormap (blue-with-red) for the difference plots such as Figure 8, where the data are centered across zero. Examples can be found in the python matplotlib documentation: https://matplotlib.org/stable/gallery/color/colormap_reference.html

[Response]: Figures 1,2,3, 7 (previously 8) have been regenerated using perceptually uniform or diverging colormaps where necessary. We found Figure 1 in Stoelzle and Stein, 2021 (https://hess.copernicus.org/articles/25/4549/2021/) especially helpful. We humbly accept this opportunity as a long overdue lesson to learn and apply henceforth.

[Reviewer 1]: Figure 4 - Suggest using a logarthmic y-scale for the spectral [RTAerr] plot. This would still make it obvious at the AIRS error spectrum is much smaller than the CrIS FSR error spectrum, but also show the spectral shape of the AIRS error spectrum which is of interest in its own right.

[Response]: We agree with the Reviewer on this point in principle, but after careful consideration decided against such a change to this figure. Instead, we added the following two sentences to the discussion of Figure 4.

"Of importance here is the large disparity in [RTAerr] across the Aqua and SNPP configurations, not a detailed depiction of each [RTAerr] individually."

[Reviewer 1]: Figure 6 - the overplotted lines are very unclear, particularly in panel (b) with 4 lines over top each other. Instead of plotting the range of variation as a horizontal error bar, I recommend plotting the range with either thin dotted lines, or use a filled semitransparent polygon (see Figure 4 of https://amt.copernicus.org/articles/17/6223/2024/). Either of those options would remove the horizontal line of the error bar, which is cluttering the graphs.

[Response]: We accept the Reviewer's point, but disagree on the course suggested. Semitransparent overlapping polygons create their own type of confusion. The interpretation of graphic figures is always subjective, and sometimes clutter is unavoidable. We would like to make the case for keeping Figure 6 as is, since the overlapping of profiles in panel b is exactly the point we make. A close analysis of the differences would distract from the overall goal in this paper.

[Reviewer 1]: Finally, I suggest splitting apart (b) and (c) into multiple subplots - in each case, plot only two cases together, e.g., plot together the "V2" data (as the baseline performance) along with one of the test configurations. Also, the range of variation is plotted as +/- 1 Std Dev, but I might expect these distributions could be very skewed. If switching to using percentiles to mark the range reveals skewed distributions, then those should be used instead of the standard deviation.

[Response]: We have considered the Reviewer's recommendations and decided to keep Figure 6 as is. Figure 6b and 6c show relative differences in the experiments. Of importance is the low variance about the mean AK profile for V2.0 (red) relative to the other configurations, which is made clear in the figure as it stands.

Grammatical corrections
* * *
[Reviewer 1]: Line 173: "discreet" -> "discrete"

[Response]: Fixed.

[Reviewer 1]: Line 185: "AK" - I don't think this was defined as "Averaging Kernel" yet - it looks like this described around line 502.

[Response]: The acronym, AK, is now defined in the abstract and standardized throughout the paper.

[Reviewer 1]: Line 211: "ADIFF > 0" - an absolute value is implied here, I think. I would make it explicit to be more clear ("|ADIFF| > 0")

[Response]: Fixed.

[Reviewer 1]: Line 254: "Cloud clearing is spatially linear, so cloud-cleared radiances are retrieved first." I would think the primarily rationale is that the radiative effect of clouds are by far the largest control on the measured radiance, and that the spectral perturbation impacts most of the spectral range.
Line 257: "(iii) [h2o_vap] vapor is highly non-linear, but it can be retrieved with accuracy once [air_temp] is known". Similar to above - doesn't Water vapor need to be the first concentration retrieval, because it is the largest perturbation (after clouds and temp profile), and it covers so much of the spectrum?

[Response]: Yes, these are both a true statements. We have changed the text according to the reviewer suggestions as follows: "(i) Clouds are by far the largest spectral signature and affects most of the measured spectral range, so cloud-cleared radiances are retrieved first…(iii) [h2o_vap] is the largest perturbation after clouds and [air_temp]. Water vapor is also pervasive in the whole spectrum. This is why we retrieve [h2o_vap] after the first [air_temp] retrieval and why it precedes all the other gases."

[Reviewer 1]: Line 286: "can primarily be attributed to the time difference in observation between the two satellites". Please remind the reader what the approximate time difference is for the data under analysis (2016).

[Response]: We added the following sentence: "While Aqua and SNPP both have local overpass times of 01:30 am/pm at nadir, the difference in their orbital altitude means their measurements are offset by a few minutes"

[Reviewer 1]: Line 291: Maybe missing some words here, I think it needs to say: "...while the AIRS FOVs maintain their alignment with the cross-track scan direction irrespective of view angle."

[Response]: Implemented as suggested, thanks.

[Reviewer 1]: Line 320: "... the fact that AIRS requires a frequency correction " Is there a reference for this? I am unaware of this effect, or what it means.

[Response]: We added a note at the end of this sentence that refers the reader to the Supplement. It is beyond the scope of our work described in this paper to go into any detail regarding instrument design. Instead, we make the reader aware of the actions we need to take for the sake of a continuous Level-2 retrieval record.

[Reviewer 1]: Line 335: "... the two MERRA2 reanalysis fields on either side of the measurement in space and time, and interpolating to the exact location.". Is this linear interpolation? (please state in the text) - if it is something more complicated than linear interpolation, is there a reference available?

[Response]: We now state "linearly interpolate"

[Reviewer 1]: Line 346 : "AK,s" -> "AK's"

[Response]: Fixed as "AKs" or the sake of consistency with other mentions of plural averaging kernels.

[Reviewer 1]: Line 367: "degreasing" -> "decreasing"

[Response]: Fixed.

[Reviewer 1]: Line 429: " ...and thus remove a significant portion of the geophysical noise otherwise present in the CrIS interferograms". I think this should refer to measurement noise, not geophysical noise? I am not sure how apodization of the spectrum would change the geophysical noise.

[Response]: No. We mean geophysical noise, not measurement noise. What we mean by geophysical noise is the noise in a given channel that is induced by geophysical parameters that are not known well and/or held constant for a given step but have some error - i.e., what we are calling interference.

The more localized a channel instrument line function (ILS) is, the more likely it is to be only sensitive to a few parameters (ideally one parameter). The broader the ILS is the more likely it

will be simultaneously sensitive to a large number of geophysical parameters. For CrIS, the Hamming apodized ILS has a full-width-half-maximum (FWHM) of ~1.35 cm-1 and the small residual side-lobes drop off extremely fast - so Hamming apodized CrIS channels are extremely localized and similar to AIRS. The unapodized ILS has very large side-lobes (-21%, +12%, -9%, +7%, ...) that drop off very slowly. Spectral lines 20 FWHM away still contribute ~1% to the signal. At the CrIS noise level, channels in mid-band have significant sensitivity to the signals in every other channel within that band. Thus, unapodized CrIS convolves the geophysical signals of an entire band into each and every channel of that band. If one knew the geophysical parameters a-priori they can be separated, but therein lies the problem. Apodization effectively isolates the spectrum, and therefore, the geophysical signals to a narrow ILS.

So imagine a channel that has a very localized ILS (e.g., CrIS Hamming apodized) versus the same unapodized channel that have strong side-lobes that extend 100's of channels. Now imagine we are doing a water vapor or CH4 retrieval and in the side-lobes are very strong signals due to volcanic SO2 of ~30 K at ~1360 cm-1. An apodized channel would have no sensitivity to volcanic SO2, whereas an unapodized channel would have a strong SO2 signal (relatively speaking) and almost every water vapor or CH4 retrieval would thus be corrupted with an SO2 signal. We would be forced to know SO2 a-priori to retrieval of water vapor at every scene, which is not possible. An alternative approach would be to retrieve SO2 simultaneously with water vapor and CH4, but that would introduce a new suite of complex errors.

[Reviewer 1]: Footnote in Table 1 (around line 444) - I think this should refer to Figure 5, not Figure 4.

[Response]: Fixed.

[Reviewer 1]: Line 494: "SNR vary" -> "SNR varies"

[Response]: Fixed.

[Reviewer 1]: Line 535: Should this be B_max = 0.2 ? (Not 2.0)

[Response]: Fixed.

[Reviewer 1]: Line 666: "now absent" -> "not absent"

[Response]: Fixed.

[Reviewer 1]: Line 667: "systematically propagating" -> "systematic propagation"

[Response]: Fixed.
* * *
**Author response to Reviewer 2**

The paper introduces new settings related to the CLIMCAPS retrieval for different instruments and satellites. It is well-structured and provides detailed explanations, but some sections are overly lengthy and could be revised for conciseness. Additionally, there are several minor corrections that need to be addressed.

[Response]: We are grateful to the reviewer for their review, which we address in-line below.

Major revisions:
* * *
[Reviewer 2]: The abstract provides extensive details about the instruments, satellites, and the CLIMCAPS system, but it lacks a focus on the results and the novelty of the paper. The results and their implications are only briefly summarized. Please consider a more balanced abstract.

[Response]: We rewrote the abstract as follows.

"*The Community Long-term Infrared Microwave Combined Atmospheric Product System (CLIMCAPS) characterizes the atmospheric state as vertical profiles of temperature, water vapor, $CO_2$, CO, $CH_4$, $O_3$, $HNO_3$ and $N_2O$, together with a suite of Earth surface and cloud properties. The CLIMCAPS record spans more than two decades (2002–present) because it utilizes measurements from a series of hyperspectral infrared sounders on different satellite platforms. In this paper, we take a step-wise approach to diagnosing the CLIMCAPS V2 system with the goal of identifying which Bayesian retrieval components to improve for a future V3 release. CLIMCAPS is based on the NASA (National Aeronautics and Space Administration) heritage retrieval approach, and is the first system to extend the Aqua record with sounders on next-generation platforms in the same orbit. With the baseline quality of CLIMCAPS V2 soundings well-established, the objective of a V3 upgrade is to improve product continuity across the different instruments and platforms for the sake of a seamless global record of atmospheric soundings. We demonstrate how the retrieval averaging kernels (AKs) are key metrics in diagnosing a multi-instrument systems such as CLIMCAPS, and conclude with the recommendation to upgrade the channel subsets and radiative transfer model error spectrum used in defining the Bayesian measurement error covariance matrix*".

[Reviewer 2]: The introduction is too long, spanning about three pages. It contains excessive background information, such as discussions on MODIS, VIIRS, and CERES, which are not used in this paper and are therefore irrelevant. The introduction could be condensed to focus more on the study's specific goals and context.

[Response]: Agreed. We have removed excessive content to focus the discussion on CLIMCAPS.

[Reviewer 2]: Section 2 is also too lengthy. For example, there is a detailed explanation of NUCAPS and AST-Aqua V7, which, while useful for comparison, could be summarized more. The discussion of dynamic regularization and SVD could also be more concise.

[Response]: With fresh eyes, we see how the mention of NUCAPS and AST-Aqua V7 distracts from the discussion, especially since we never show any results from those systems. For this

revision, we removed all comparisons to NUCAPS and AST-Aqua V7 for the sake of clarity, and mention NUCAPS for the first time in the Conclusion, since the results reported in this paper may be relevant to NUCAPS in future.

[Reviewer 2]: Section 3, Experimental Design, also repeats certain concepts multiple times. For instance, the explanations about the sequential retrieval process and the covariance matrix are repeated unnecessarily.

[Response]: We have reworked this section for the sake of focus and clarity.

Minor corrections:
* * *
[Reviewer 2]: Line 63: NASA AST: what is AST stand for? It was not mentioned in the paper.

[Response]: We have removed all mention of "AST" (AIRS science team) in this paper.

[Reviewer 2]: Line 117: What is NUCAPS stand for?

[Response]: Our mention of NUCAPS in Line 117 was removed during the rewrite. Upon first mention of NUCAPS in the Conclusion, we include the acronym description.

[Reviewer 2]: Line 184: The acronym AKs is introduced for the first time in line 184, but its meaning is only explained later in Section 2.3. and again later in Fig.6 (section 4) was mentioned about Averaging Kernel. Please consider this inconsistency.

[Response]: Fixed throughout.

[Reviewer 2]: Line 274: The same story for „FOR", it was defined for the 1st time in line 274 but it was used several time before it for example in line 210 , …

[Response]: Fixed.